# Vegetation change across the Drake Passage region linked to late Eocene cooling and glacial disturbance after the Eocene-Oligocene Transition

Nick Thompson[1], Ulrich Salzmann[1], Adrián López-Quirós[2,3], Peter K. Bijl[4], Frida S. Hoem[4], Johan Etourneau[3], Marie-Alexandrine Sicre[5], Sabine Roignant[6], Emma Hocking[1], Michael Amoo[1], Carlota Escutia[3]

[1]Department of Geography and Environmental Sciences, Northumbria University, Newcastle upon Tyne, UK.
[2]Department of Geoscience, Aarhus University, Høegh-Guldbergs Gade 2, 8000, Aarhus C, Denmark.
[3]Instituto Andaluz de Ciencias de la Tierra, CSIC-Universidad de Granada, Granada, Spain.
[4]Department of Earth Sciences, Utrecht University, Utrecht, The Netherlands.
[5]Sorbonne Universites (UPMC, Univ. Paris 06)-CNRS-IRD-MNHN, LOCEAN Laboratory, Paris, France.
[6]Institut Universitaire Europeen de la Mer, Plouzane, France.

*Correspondence to*: Nick Thompson (alasdair.thompson@northumbria.ac.uk)

**Abstract.** The role and climatic impact of the opening of the Drake Passage and how it affected both marine and terrestrial environments across the Eocene-Oligocene Transition (EOT ~34 Ma) period remains poorly understood. Here we present new terrestrial palynomorph data compared with recently compiled lipid biomarker (*n*-alkane) data from Ocean Drilling Program (ODP) Leg 113 Site 696 drilled on the margin of the South Orkney Microcontinent (SOM) in the Weddell Sea, to investigate changes in terrestrial environments and paleoclimate across the late Eocene and early Oligocene (~37.6-32.2 Ma). Early late Eocene floras and sporomorph-based climate estimates reveal *Nothofagus*-dominated forests growing under wet temperate conditions, with mean annual temperature (MAT) and precipitation (MAP) around 12°C and 1802 mm, respectively. A phase of latest Eocene terrestrial cooling at 35.5 Ma reveals a decrease in MAT by around 1.4°C possibly linked to the opening of the Powell Basin. This is followed by an increase in reworked Mesozoic sporomorphs together with sedimentological evidence indicating ice expansion to coastal and shelf areas approximately 34.1 million years ago. However, major changes to the terrestrial vegetation at Site 696 did not take place until the early Oligocene, where there is a distinct expansion of gymnosperms and cryptogams accompanied by a rapid increase in taxa diversity and a shift in terrestrial biomarkers reflecting a change from temperate forests to cool temperate forests following 33.5 Ma. This surprising expansion of gymnosperms and cryptogams is suggested to be linked to environmental disturbance caused by repeat glacial expansion and retreat, which facilitated the proliferation of conifers and ferns. The timing of glacial onset at site 696 is linked to the global cooling at the EOT and latest Eocene regional cooling cannot directly be linked to the observed vegetation changes. Therefore, our vegetation record provides further evidence that the opening of the Drake Passage and Antarctic glaciation were not contemporaneous, although stepwise cooling in response to the opening of ocean gateways surrounding the Antarctic continent may have occurred prior to the EOT.

## 1. Introduction

The Cenozoic progression from greenhouse to icehouse climate conditions was accompanied by the establishment of the Antarctic ice sheet around the Eocene-Oligocene Transition (EOT 34.44-33.65 Ma; e.g., Hutchinson et al., 2021). This change in Earth climate state is evidenced by a prominent excursion in oxygen isotope ratios from marine biogenic calcite (e.g., Zachos et al., 2001, 2008; Westerhold et al., 2020) during the Earliest Oligocene Oxygen Isotope Step (EOIS ~33.65Ma; Hutchinson et al., 2021). The possible causes of the onset of Antarctic glaciation are poorly understood and ambiguity remains as to

whether a single or combination of factors and feedbacks drove the Cenozoic climate transition (e.g., DeConto and Pollard, 2003; Coxall and Pearson, 2007). In particular, large uncertainties remain over the role of the opening and deepening of the Drake Passage on the development of the Antarctic Circumpolar Current (ACC), and how this affected both marine and terrestrial environments (Scher and Martin, 2008; Houben et al., 2019; Lauretano et al., 2021). Today ocean currents and the ACC exert a strong influence on the Earth's climate system in the global distribution of heat, nutrients, salt and carbon, as well

as in the gas exchange between the atmosphere and the ocean (Cox, 1989; Bryden and Imawaki, 2001; Anderson et al., 2009; Sarkar et al., 2019). In particular the ACC facilitates the thermal isolation of Antarctica from subtropical surface heat through the isopycnal tilt of its water masses, acting to stabilise the Antarctic ice sheet (Martinson, 2012; Sarkar et al., 2019). Given that unabated anthropogenic warming is expected to cause a poleward shift of the ACC and potentially weaken thermohaline circulation (Zhang and Delworth, 2005) this study forms part of a wider need to fully understand the Earth climate system in

order to better predict future stability of the Antarctic ice sheet.

  A major obstacle in understanding the role of the opening Drake Passage and ocean currents in Cenozoic climate change has been the lack of well-dated continuous records spanning the EOT from the region. Here we present new terrestrial palynomorph data from the EOT recovered in the Ocean Drilling Program (ODP) Leg 113 Site 696 Hole B (herein referred to as Site 696),

containing a well recovered EOT section (~37.6-32.2 Ma; Houben et al., 2013, 2019). Vegetation composition, structure and diversity patterns are reconstructed along with sporomorph-based quantitative climate estimates in order to explore the timing and nature of vegetation and climate change across the northern Antarctic Peninsula region and South Orkney Microcontinent (SOM). The results are compared with recently compiled lipid biomarker (*n*-alkane) data (López-Quirós et al., 2021), and dinoflagellate cyst data (Houben et al., 2013) to better understand shifts in marine as well as terrestrial environments and the

source of terrestrial versus aquatic organic matter. Our results reveal new insights into the timing of terrestrial climate cooling in the region and glacial onset on Antarctica across the EOT.

## 2. Previous Geochemical Analyses

 The following section will focus on the interpretation of lipid biomarker (*n*-alkane) and stable isotope data from Site 696 by López-Quirós et al. (2021). For a full description of geochemical methods see López-Quirós et al. (2021).

## 2.1 Lipid biomarkers (*n*-alkanes)

The distribution of *n*-alkanes in sediments can be assessed on the basis of carbon chain length in order to determine potential biological sources (Cranwell, 1973; Rieley et al., 1991; Bi et al., 2005; Duncan et al., 2019; López-Quirós et al., 2021). Algae and bacteria typically produce shorter chain lengths ($C_{12}$-$C_{22}$; Clark and Blumer, 1967; Han and Calvin, 1969; Cranwell et al., 1987; Grimalt and Albaigés, 1987; Duncan et al., 2019), while aquatic plants and *Sphagnum* mosses are characterised by enhanced production of $C_{23}$ to $C_{25}$ chain lengths (Baas et al., 2000; Ficken et al., 2000; Pancost et al., 2002; Bingham et al., 2010; Duncan et al., 2019). Long chain *n*-alkanes ($C_{25}$ and higher) are most abundantly produced by terrestrial higher plants (Eglinton and Hamilton, 1963; Duncan et al., 2019). Therefore, medium to long chain *n*-alkane ($C_{23}$-$C_{31}$) distributions can provide details about the origin of organic matter in sediments, differentiating between terrigenous and marine, providing information about palaeovegetation and palaeoclimate (Meyers et al., 1997; Ficken et al., 2000; Schefuß et al., 2003; Vogts et al., 2009; Duncan et al., 2019; López-Quirós et al., 2021).

### 2.1.1 ACL (Average Chain Length n-alkane index)

Variations in the ACL index through time can be used as a proxy of terrestrial organic matter inputs and can also provide information on changes in climate (Collister et al., 1994; Rommerskirchen et al., 2006; Mahiques et al., 2017; Duncan et al., 2019; López-Quirós et al., 2021). Plants produce higher ACLs in warmer, tropical regions, whilst lower ACLs are generally observed from cooler climates (Poynter et al., 1989; Sicre and Peltzer, 2004; Jeng, 2006; Vogts et al., 2009; Bush and McInerney, 2015; Duncan et al., 2019; López-Quirós et al., 2021). Studies have also suggested plants synthesise longer *n*-alkanes in more arid environments providing plants with a more efficient wax coating to restrict water loss (e.g., Kolattukudy et al., 1976; Schefuß et al., 2003; Calvo et al., 2004; Zhou et al., 2005; Moossen et al., 2015; Jalali et al., 2017, 2018), indicating aridity has a strong control on ACL and that ACL index values decrease under wetter conditions (Duncan et al., 2019; López-Quirós et al., 2021). At Site 696 ACL values display an upward decreasing trend (Fig. 5; López-Quirós et al., 2021). Higher ACLs indicate a mixed unput from higher land-pants generally synthesized under warmer climate conditions (Jeng, 2006; Vogts et al., 2009; Bush and McInerney, 2015).

### 2.1.2 $P_{aq}$ (Aquatic Plant n-alkane index)

The $P_{aq}$ index provides an approximate measure of the relative sedimentary contribution of submerged and floating aquatic macrophytes relative to emergent and terrestrial vegetation (Ficken et al., 2000; López-Quirós et al., 2021). *Sphagnum* mosses also have a molecular distribution similar to submerged and floating macrophytes, showing enhanced production of $C_{23}$ and/or $C_{25}$ (Baas et al., 2000; Nott et al. 2000; Nichols et al., 2006; Duncan et al., 2019). Therefore, the $P_{aq}$ index reflects the input from *Sphagnum* and aquatic plants versus terrestrial vegetation. At the study site $P_{aq}$ values <0.23 indicate a dominance of terrestrial plant waxes, while higher values of 0.48 to 0.49 imply an enhanced contribution of enhanced submerged and floating, and/or *Sphagnum* moss (Fig. 5; López-Quirós et al., 2021).

### 2.1.3 TI (Terrestrial n-alkanes index)

The TI index is based on the assumption that inputs from photosynthetic algae and bacteria are characterised by short-chain *n*-alkanes ($C_{12}$-$C_{22}$; Clark and Blumer, 1967; Han and Calvin, 1969; Cranwell et al., 1987; Grimalt and Albaigés, 1987; Duncan et al., 2019) compared to higher land plants rich in $C_{27}$, $C_{29}$ and $C_{31}$ (Bourbonniere and Meyers, 1996; Mahiques et al., 2017; López-Quirós et al., 2021). The TI index is calculated as a ratio over the Total Organic Carbon (TOC; Mahiques et al., 2017). At Site 696 higher values of TI characterize a greater input of terrestrial plant-derived organic matter (Fig. 5; Mahiques et al., 2017; López-Quirós et al., 2021).

### 2.2 TOC (Total Organic Carbon) and TN (Total Nitrogen)

The ratio of total organic carbon to total nitrogen (TOC to TN) can be used to distinguish the sources of organic material, which is helpful in reconstructing the evolution on environments (Sampei and Matsumoto, 2001; Perdue and Koprivnjak, 2007). TOC represents the organic fraction preserved in sediments and can be used to help distinguish between marine and terrestrial sources of organic matter, depositional conditions and organic matter production (Calvert and Pedersen, 1993; Meyers and Ishiwatari, 1993; Avramidis et al., 2014, 2015). Analysis of sediments from Site 696 reveal a significant positive relationship ($R>0.9$; López-Quirós et al., 2021) between TOC and TN. In addition, close correspondence between the two proxy records suggests TOC and TN reflect the same bulk organic matter source. C/N ratio values from Site 696 indicate a mixture of marine- and terrestrial-derived sources (López-Quirós et al., 2021), consistent with the presence of both marine and terrestrial palynomorphs.

Organic matter in marine sediments is mainly derived from the decomposition of plants, animals and most importantly plankton (Avramidis et al., 2015). High planktonic primary production and zooplankton grazing results in increased export of organic matter through the water column to the sea floor supporting increased preservation of organic carbon in sediments. Although the C/N ratios have been interpreted to essentially be equal to the weight ratio of $C_{org.}$ to organic nitrogen (i.e., $C/N_{org.}$ ratio), the presence of inorganic nitrogen measured within TN has led some researchers to point out that a relatively high $N_{inorg.}$ could affect the C/N ratio (Müller, 1977; Sampei and Matsumoto, 2001). Therefore, TOC may be a better indication for palaeoproductivity despite dependence on degradation and thus the residence time in the water column (Sarnthein et al. 1988; Lyle et al. 1988; Berger and Herguera 1992; Freudenthal et al. 2002; Jahn et al. 2003; Luo et al., 2013; Frihmat et al., 2015). However, organic carbon burial is also affected by redox conditions, and terrigenous detrital matter influx also exert a control and should be taken into account when interpreting TOC in terms of palaeoproductivity (Luo et al., 2013). At Site 696 increased levels of TOC coincide with higher abundance of heterotrophic dinoflagellate cysts (Houben et al., 2013) and may be used to support the notion of high marine palaeoproductivity.

## 3. Materials and Methods

Site 696, hole B was drilled on the south-eastern margin of the SOM (Fig.1; latitude: 61°50.959′S, longitude: 42°55.996′W) at 650m water depth, as part of ODP Leg 113 in 1987 (Barker et al., 1988). The recovered section consists of late Eocene to Quaternary hemipelagic (214-0 mbsf), diatomaceous (530-214 mbsf), and terrigenous (645.4-530 mbsf) sediments (Barker et al., 1988; Wei and Wise, 1990; Gersonde and Burckle, 1990; López-Quirós et al., 2019, 2020, 2021) and is divided into seven lithological units (I-VII), primarily based on composition and diagenetic maturity of sediments (Fig. 2; Barker et al., 1988). This study focuses on pollen and spores recovered from the terrigenous unit VII (cores 113-696B-62R through 113-696B-53R; Fig. 2) interpreted shipboard to be deposited in a shallow marine shelf environment (Barker et al., 1988). Age-control based primarily on the presence of calcareous nannofossils (Wei and Wise, 1990 *sensu* Villa et al., 2008; and a revised dinoflagellate cysts age model (Houben et al., 2013, 2019), places the studied section at 33.2 to 37.6 Ma (Table 1), with sediments encompassing the EOT and EOIS event recovered between 571.5 mbsf to 569.1 mbsf (Houben et al., 2013).

A total of 35 samples from the late-middle Eocene to earliest Oligocene (643.73-520.88 mbsf) were analysed for their pollen and spore content. Raw data collected is available from the PANGEA database (Thompson et al., 2021). All palynological slides were prepared using standard chemical palynological processing techniques following the protocols at the University of Northumbria, Department of Geography and Environmental Sciences and the Laboratory of Palaeobotany and the Laboratory of Palaeobotany and Palynology of Utrecht University, published previously (e.g., Bijl et al., 2018; Riding, 2021). Samples were treated with 30% HCl overnight and cold 38 % HF to dissolve carbonates and silicates respectively. Next 30% HCl was then added to remove fluoride gels, and subsequently centrifuged, decanted and sieved using 250 μm to 10 μm sieve meshes. Residues were mounted on glass slides using glycerine jelly. Slides were analysed using a Leica DM500 and Leica DM2000 transmitted light microscopes at 200x and 1000x magnification. Where possible, counts of 300 (excluding reworked grains) sporomorphs were made. Only samples containing 50 or more in situ sporomorph grains were used for further analysis and evaluation.

Identification and taxonomic classification of sporomorphs were carried out primarily following Cookson (1950), Cookson and Pike (1954), Dettmann et al. (1990), Dettmann and Jarzen (1996), Truswell and Macphail (2009) and Raine et al. (2011). Botanic and taxonomic affinities used to identify the Nearest Living Relatives (NLR) of fossil species were established mainly after Truswell and Macphail (2009) and Raine et al. (2011) and references therein (Table 2.). Identification of reworked grains are mainly based on the age-restriction of the species, with species older than Eocene or Oligocene (e.g., Mesozoic species) being easily recognised as reworked. Consideration was also given to whether a grain was reworked based on the level of thermal maturity and its state of preservation. All palynomorphs identified as in situ are regarded as being penecontemporaneous with deposition and are included in the final calculation of sporomorph percentages. Pollen percentages were plotted using riojaPlot, based on the R package rioja (Juggins, 2020) and local zones were established using the CONISS

(Constrained Incremental Sum-of-Squares: Grimm, 1987) cluster analysis function. Sporomorph diversity was measured using both the Shannon–Wiener index and the observed number of taxa. A rarefaction method for sums of ≥50 and ≥100 grains was applied, so that the effect caused by differences in the sample size may be removed allowing the estimation of the number of sporomorph species at a constant sample size (Raup, 1975; Birks and Line, 1992). The Shannon–Wiener Index was also carried out as the second measure of sporomorph diversity accounting for species richness and evenness (Shannon, 1948; Magurran, 2013; Morris et al., 2014). Samples containing less than 50 grains were omitted from this analysis. Detrended Correspondence Analysis (DCA) was performed, with downweighting of rare species by removing pollen types whose representation is <5%. This ordination technique is used in order to evaluate ecological patterns within the data, using knowledge of the distribution of NLR and their modern environmental gradients (Correa-Metrio, 2014). Rarefaction, Shannon–Wiener and DCA were all performed using the software R for statistical computing (R Development Core Team, 2013) and the package Vegan (Oksanen et al., 2013).

### 3.1 Bioclimatic Analysis

Estimates for terrestrial mean annual temperature (MAT), mean annual precipitation (MAP), warmest month mean temperature (WMMT) and coldest month mean temperature (CMMT) were obtained using the NLR approach in conjunction with the Probability Density Function (PDF) method. Fossil taxa used and their NLR are shown in Table 2.

Climate estimates based on the NLR approach use presence or absence data and are independent of the relative abundance of individual taxa. This makes this method ideal for sporomorph based climate estimates from marine sediments, where hydrodynamic sorting of grains may cause variations in the percentages of individual taxa (Arias, 2015), and also helps reduce taphonomic biases (Klages et al., 2020). However, the assumption that modern species and their climate requirements have remained unchanged throughout geological time represents one of the biggest weaknesses of the NLR approach. This uncertainty inevitably increases the further back in the geological record (Hollis et al., 2019). It should also be noted that the modern distribution of species may be a function of either its past climate or biogeographic history (Reichgelt et al., 2016; Willard et al., 2019). Nevertheless, temperature estimates derived from the NLR approach are often in agreement with those from other botanical methods and geochemical proxies, such as the Climate Leaf Analysis Multivariate Program (CLAMP) and leaf margin analysis (e.g., Kennedy, 2003; Uhl et al., 2003; Ballantyne et al., 2010; Pross et al., 2012; Kennedy et al., 2014; Pound and Salzmann, 2017; Willard et al., 2019) providing a certain level of confidence (Klages et al., 2020; Pross et al. 2012).

The PDF method is used to statistically constrain the most likely climate co-occurrence window for an assemblage (Harbert and Nixon, 2015; Willard et al., 2019; Klages et al., 2020). The bioclimatic envelope for each NLR was identified by cross plotting the modern distribution from the Global Biodiversity Information Facility (GBIF; GBIF, 2021) with the gridded WorldCLIM (Fick and Hijmans, 2017) climate surface data using the dismo package (Hijmans et al., 2017) in R. Some taxa

were grouped at the family level because of their potentially ambiguous climatic affinity. This includes (1) pollen taxa affiliated with the modern-day genus *Microcachrys*, of which *Microcachrys tetragona* is the sole species, on the basis that *M. tetragona* is only found in a particular location in Tasmania, Australia, under narrow climatic and environmental conditions which are likely not representative of this once widespread genus; and (3) the pollen taxa *Peninsulapollis gillii*, which has links to the modern genus *Beauprea* now also endemic to New Caledonia. In these cases, Podocarpaceae and Proteaceae were used, respectively, rather than the genus or species as the NLR.

## 4. Results

The recovery of palynomorphs is good throughout the section. Of the 34 samples analysed 5 do not contain a sufficient amount of sporomorphs and were discarded from further analysis. In total 74 pollen taxa (58 angiosperms and 16 gymnosperms), 24 spores and 1 sporomorph of unknown affiliation were identified (excluding reworked and unidentified sporomorphs), containing 54 genera. The stratigraphic distribution and relative abundance of major taxa groups is shown in Fig. 3. Pollen affiliated with the modern-day genus *Nothofagus* are the most abundant throughout the section, with pollen taxa belonging to the *Nothofagidites lachlaniae* complex, undifferentiated *Nothofagidites* spp., *Nothofagidites rocaensis* and the *Nothofagidites brachyspinulosus* complex being the largest groups. Other major pollen and spore taxa, in order of decreasing abundance include, undifferentiated *Podocarpidites* spp., undifferentiated *Retitriletes/Lycopodiacidites* spp., *Podocarpidites cf. exiguus*, pollen belonging to the *Podocarpidites marwickii/ellipticus* complex, *Cyathidites minor* and *Phyllocladidites mawsonii*, which occur commonly throughout the Eocene and Oligocene sections.

Based on the results of CONISS ordination the succession is divided into 2 main zones (I and II; Fig. 3). In addition, Zone I is further subdivided (Ia Ib), based on the abundance and presence of key taxa. The results of rarefaction and DCA analysis along with the diversity indices results also show a good distinction between Zones I and II.

### 4.1 Zone I, 37.6-33.6 Ma (643.73-568.82 mbsf)

Zone I comprises of 18 samples (62R 6W 142-144 to 55R 1W 62-64). Based on the age models of Wei and Wise (1990) and Houben et al. (2013, 2019) and linear extrapolation, the lowermost 16 samples are placed in the Eocene, while the uppermost 2 samples are placed into the earliest Oligocene (37.6 Ma to ~33.6 Ma). Quantitatively, Zone I is typified by relatively low numbers of sporomorph species and low diversity. Based on rarefaction analysis, the average number of sporomorph species per sample is $13.28 \pm 1.05$ (mean ± SD) at a count of 50 grains. Low levels of diversity are confirmed by the Shannon diversity indices (H), which indicates an average of $1.79 \pm 0.06$.

The overall Zone I assemblage is dominated by the southern beech, *Nothofagus* (pollen taxa: *Nothofagidites*). On average *Nothofagidites* pollen accounts for 79.0% of all non-reworked taxa and 95.0% of all angiosperm taxa. Taxa belonging to the *Nothofagidites lachlaniae* complex (subgenus: *Fuscospora*) are the most abundant followed by undifferentiated *Nothofagus*

spp. sporomorphs and *N. rocanensis* (subgenus: *Nothofagus*). Other angiosperm pollen (non-*Nothofagidites*) is rare, making up about 4.2% of the non-reworked sporomorph assemblage in Zone 1. Of the non-*Nothofagus* angiosperm taxa the most abundantly occurring, in order of decreasing abundance, include *Proteacidites* (NLR: Proteaceae), *Tricolpites* (Dicotyledonae), *Liliacidites intermedius* (Liliaceae) and *Lateropora glabra* (*Freycinetia*). Other less common angiosperms are typically only represented by one or two occurrences. The second most abundant group are gymnosperms, which account for 10.6% of all non-reworked taxa. Predominantly gymnosperms are represented by the pollen taxa (in order of abundance) *Podocarpidites*, *Phyllocladidites*, *Trichotomosulcites subgranulatus* (all Podocarpaceae), undifferentiated *Podocarpidites* spp. (*Podocarpus*) and *Araucariacites australis* (Araucariaceae). Many of these are likely to belong to *Podocarpidites* however folding of the grains has made further identification impossible. Cryptogams account for 6.23% of non-reworked taxa in Zone 1 and include both ferns and mosses. Abundantly occurring cryptogam spores include taxa belonging to the *Retitriletes/Lycopodiacidites* spp. complex (Lycopodiaceae), *Cyathidites* (Cyatheaceae), *Ischyosporites gremius* (Filicopsida) and *Coptospora archangelskyi* (*Conostomum*).

### 4.1.1 Subzone Ia 37.6-35.5 Ma (643.73-597.66 mbsf)

The Subzone Ia assemblage is unique in that *Arecipites* spp. (Arecaceae), *Beaupreaidites* (Beauprea) and *Myrtaceidites cf. mesonesus* (Myrtaceae), all warmth loving taxa whose NLRs predominantly have a tropical and subtropical distribution, especially in the Pacific, Southeast Asia and New Caledonia, and only occur in this subzone. *Ericipites cf. scabratus* (Ericaceae), *Chenopodipollis cf. chenopodiaceoides* (Chenopodioideae), *Polypodiisporites cf. radiatus* (*Davallia*), *Podosporites parvus* (Podocarpaceae) and *Tricolpites cf. asperamarginis* (extinct clade) are also unique to Subzone Ia. In addition, pollen taxa belonging to the genus *Podocarpidites* are more abundant throughout Subzone Ia compared to Subzone Ib, in particular taxa belonging to the *Podocarpidites marwickii/ellipticus* complex and *P. cf. exiguus* (both *Podocarpus*). Furthermore, taxa belonging to the *Nothofagidites asperus* complex (subgenus: *Lophozonia*), *Microcachryidites antarcticus*, *Trichotomosulcites subgranulatus* (both Podocarpaceae), *Gleicheniidites* (Gleicheniaceae) and *Ischyosporites* (Filicopsida) are also more abundant in Subzone Ia in comparison to Subzone Ib. Other rare taxa such as *Lymingtonia cf. cenozoica* (Nyctaginaceae), *Myrtaceidites* spp. (Myrtaceae), *Proteacidites tuberculatus* (Proteaceae) and *Ceratosporites cf. equalis* (Selaginellaceae) also only occur in Subzone Ia of Zone I but are represented by one or two specimens. Sporomorph-based climate reconstructions reveal significantly higher temperatures within Subzone Ia compared to Subzone Ib, with an interval of latest Eocene cooling occurring around 35.5 Ma. MAT ranges from 10.5°C to 15.3°C and MAP ranges from 1580mm to 2005mm, with an average of 12°C and 1802mm respectively for Subzone Ia (Fig. 4).

### 4.1.2 Subzone Ib 35.0Ma-33.6 Ma (588.25-568.82 mbsf)

Subzone Ib records the loss of thermophilic plant types (Arecaceae, Beauprea and Myrtaceae) that are only found within Subzone Ia and a decrease in the abundance of Podocarpaceae. In comparison to Subzone Ia, taxa belonging to Proteaceae are

more abundant within Subzone Ib, in particular the pollen taxa belonging to the *Proteacidites parvus/pseudomoides* complex, *P. cf. Scabratriporites* spp. and *P. tenuiexinus* (all Proteaceae). Other pollen taxa that increase in abundance and frequency within Subzone Ib of Zone I include *Liliacidites intermedius* (Liliaceae), *Tricolporites cf. scabratus* (extinct clade), Coptospora archangelskyi (*Conostomum*) and *Retitriletes/Lycopodiacidites* spp. (*Lycopodium*). Other rare taxa are also unique to Subzone

Ib of Zone I and are represented by one of two occurrences. These include *Clavatipollenites ascarinoides* (*Ascarina*), *Ligulifloridites* (Asteraceae), Parsonsidites psilatus (Malvaceae), *Proteacidites cf. amolosexinus*, *P. cf. Lewalanipollis trycheros*, *P. scaboratus*, *P. spiniferus* (all Proteaceae), *Sparganiaceaepollenites barungensis* (*Sparganium*) (*Tricolpites cf. brevicolpus*, *T. reticulatus* (both extinct clade) and *Camarozonosporites* sp. (Lycopsida). Sporomorph-based climate estimates reveal MAT between 10.1°C and 11.7°C and MAP between 1499mm and 2042mm, with an average of 10.7°C and 1706mm

respectively for Subzone Ib (Fig. 4).

### 4.2 Zone II, ca. 33.5-32.2 Ma (563.38-549.70 mbsf)

The 11 samples of Zone II (53R 1W 80-82 to 54R 3W 38-41) are assigned an Oligocene age. Zone II records a significant increase in gymnosperms and cryptogams, accompanied by a rapid rise in taxa diversity between ca. 33.5 and 32 Ma and a contemporaneous increase in reworked Mesozoic sporomorphs (Fig. 3). Based on the results of rarefaction analysis the average

number of sporomorph species for a count size of 50 grains is $19.63 \pm 2.00$. The results of the Shannon diversity index are between 1.97 and 2.12, with an average of $2.06 \pm 0.05$.

The Zone II sporomorph assemblages shows a significant decrease in *Nothofagus* compared to Zone I. *Nothofagus* pollen make up 51.2% of all non-reworked taxa and 89.9% of all angiosperm taxa in Zone II. Pollen taxa belonging to the *Nothofagidites*

*lachlaniae* complex (subgenus: *Fuscospora*) remain the most abundant, followed by *N. rocanensis* (subgenus: *Nothofagus*), with undifferentiated *Nothofagidites* spp. sporomorphs also making a valuable contribution. Other pollen taxa belonging to *Nothofagus* are less abundant and represented by only a few occurrences. Although a slight increase in other angiosperms (non-*Nothofagus*) occurs in Zone II they remain the smallest botanical group, representing just 5.8% of all non-reworked sporomorphs. In order of abundance, from most to least abundant, significant non-*Nothofagus* angiosperm taxa include

*Proteacidites*, *Tricolpites*, *Myricipites harrisii* (Casuarinaceae) and *Peninsulapollis gillii* (Proteaceae). Additional angiosperm taxa are typically represented by one or two occurrences (e.g., *Chenopodipollis chenopodiaceoides*). Gymnosperms remain the second most abundant botanical group, but their abundance has increased markedly, representing 28.3% of all non-reworked sporomorphs in Zone II. The gymnosperm assemblage remains dominated by *Podocarpidites* and *Phyllocladidites*, which are the two most common gymnosperm taxa respectively. However, other changes in the gymnosperm pollen include

*Dilwynites* (*Wollemia*) which increasing in frequency and abundance, along with *Alisporites cf. australis* (Gymnospermopsida), *Microcachryidites antarcticus*, *Podosporites*, *Trichotomosulcites subgranulatus* (all Podocarpaceae) and undifferentiated *Podocarpidites* spp. (*Podocarpus*), among others. Sporomorph-based climate estimates provide no evidence for abrupt cooling at the Eocene/Oligocene boundary. Within the early Oligocene Zone II MATs are between 10.4°C

to 12.9°C and MAP ranges from 1571mm to 1951mm a year, with an average of 11.2°C and 1715mm respectively (Fig. 4). These results indicate a slight increase in both temperature and precipitation compared to the latest Eocene Subzone Ib.

## 5. Discussion

### 5.1 Sediment Transport and Provenance

The late Eocene terrestrial vegetation assemblage from Site 696 shares a number of similarities with Antarctic Peninsula palaeofloras of similar ages (e.g., Warny and Askin 2011b; Warny et al., 2019). Both Site 696 and Antarctic Peninsula late Eocene assemblages are dominated by *Nothofagidites* pollen, predominantly those related to the modern subgenus *Fuscospora*, with secondary gymnosperms, including high frequencies of Podocarpaceae pollen. Similar angiosperm and cryptogam assemblages are also seen between the two, with angiosperms such as Proteaceae and Liliaceae, and cryptogams such as Cyatheaceae and *Sphagnum*. This finding suggests that pollen from the Antarctic Peninsula region could have been transported to the SOM during this time. Furthermore, similarities between Facies IV and nearby Seymour Island sediments, both in composition and paleogeographic setting, could suggest a related sediment source, and that the SOM was proximal enough to receive detritus from the Antarctic Peninsula (Barker et al., 1988; López-Quirós et al., 2021). However, despite these similarities, significant differences in the palaeoflora occurs between the two regions indicating the Antarctic Peninsula may not have been the primary sediment source. In agreement with previous observations by Mohr (1990) the sporomorph assemblage from Site 696 contains a greater diversity of angiosperm pollen compared to late Eocene Antarctic Peninsula palaeofloras (e.g., Anderson et al., 2011; Warny and Askin 2011b; Warny et al., 2019). This higher diversity has also been reported in southern South American Paleogene sporomorph floras (e.g., Romero and Zamaloa, 1985; Romero and Castro, 1986). In addition, the late Eocene Zone Ia assemblage (37.6-35.5 Ma) at Site 696 contains the thermophilic taxa *Arecipites* spp. (Arecaceae), *Myrtaceidites cf. mesonesus* (Myrtaceae), and *Polypodiisporites cf. radiatus* (Davallia) not recorded in coeval Antarctic Peninsula assemblages, possibly due to the more northern latitude of the SOM resulting in milder climatic conditions. Sediments may also have been supplied from the southern tip of South America (e.g., the Magallanes Basin and the Fuegian Andes; Carter et al., 2017), due to the more proximal location of the SOM to South America prior to its separation from Antarctica during the Eocene (Eagles and Jokat, 2014). However, detrital zircon ages clearly show a strong dissimilarity between Site 696 samples and South America (Carter et al., 2017). Furthermore, the occurrence of well-preserved palynomorphs and moderate to well-preserved in situ benthic foraminifera, with predominantly angular to subangular terrigenous particles, does not support the notion of long-distance transport of sediments from adjacent sources (e.g., Seymour Island and southern South America; López-Quirós et al., 2021). These observations, together with an expansion of gymnosperm conifers and cryptogams recorded during the early Oligocene (33.5-32.2 Ma) at Site 696, but absent from Antarctic Peninsula and southern South America floras (e.g., Askin et al., 1992; Anderson et al., 2011), suggest that the vegetation of the SOM was unique in character. It is therefore likely that a significant proportion of detrital material, including

sporomorphs, was likely of local origin (e.g., exposed parts of the SOM), with some input from the northern Antarctic Peninsula and possibly southern South America during the late Eocene.

The SOM and the northern Antarctic Peninsula underwent significant rifting during the late Eocene and early Oligocene (~37-30 Ma; King and Barker, 1988; Eagles and Livermore, 2002; van de Lagemaat et al., 2021), forming what would become the Powell Basin (Eagles and Livermore, 2002; Eagles and Jokat, 2014; van de Lagemaat et al., 2021; López-Quirós et al., 2021). This rifting resulted in the capture of terrigenous detritus likely from the Northern Antarctic Peninsula and exposed parts of the SOM (South Orkney Islands; Carter et al., 2017). However, throughout the latest Eocene (~35.5–34.1Ma), a decrease in the delivery of coarse terrigenous sediments and a decrease in sedimentation rates by almost half is observed as the SOM became more distal from the Antarctic Peninsula due to the opening of the proto-Powell Basin (Eagles and Livermore, 2002; López-Quirós et al., 2021). Deposition of moderately to intensely bioturbated silty mudstones across the EOT (~34.1–33.6Ma) indicate continued subsidence-related marine transgression at Site 696 (López-Quirós et al., 2021). This subsequent and continued isolation of the SOM may have resulted in Site 696 receiving a greater proportion of localised sediments from exposed parts of the SOM. This supports our suggestion that the majority of sediments supplied to Site 696 at this time were of local origin, perhaps still with some contribution from the northern Antarctic Peninsula.

Conversely however, Carter et al. (2017) suggested the majority of the late Eocene (~36.5–33.6 Ma) sediments deposited at Site 696 are not of local origin. Using detrital zircon U-Pb and apatite thermochronometry analysis these authors concluded that sand grains, featuring characteristics of ice transport, from the late Eocene Site 696 best matched sources within the Ellsworth–Whitmore Mountains, West Antarctica. Barriers to the delivery of sediment by long distance gravity flows from the margins of the southern Weddell Sea, further suggested that sediments may have been transported to the SOM by icebergs (Carter et al., 2017). In spite of this, the presence of in situ thermophilic taxa within the early-late Eocene of Site 696 (37.6-35.5 Ma) suggests mild and even ice-free conditions during this overlapping time period. Furthermore, palaeo-sea-surface temperature reconstructions (Douglas et al., 2014) indicate relatively warm conditions (~14°C), and fossil dinoflagellate cyst (Houben et al., 2013, 2019), calcareous nannofossils (Wei and Wise, 1990) and smectite-dominated clay mineralogy (Fig. 2: Robert and Maillot, 1990) support temperate depositional conditions (López-Quirós et al., 2021) not favourable for transport by ice. Unequivocal evidence for ice transport, in the form of ice-rafted debris, at Site 696 is observed within two coarse-grained mudstone intervals within a fine-grained transgressive sequence deposited around 34.1 Ma (Barker et al., 1988; López-Quirós et al., 2021). However, these intervals contain altered glaucony grains most likely sourced from shallower SOM coastal/shelf areas (López-Quirós et al., 2019, 2021). Therefore, these observations and those of this study suggest that transportation by ice from adjacent land areas (e.g., Antarctic Peninsula and Ellsworth–Whitmore Mountains) was unlikely before 34.1 Ma and that a majority of sediments transported to Site 696 are likely of local origin from exposed parts of the SOM as the Powell basin opened isolating the microcontinent from the possible sediment supply of the Antarctic Peninsula and southern Weddell Sea margins.

**5.2 Palaeoenvironment and Palaeoclimate**

### 5.2.1 Late Eocene Palaeoenvironment and Palaeoclaimte

Sediments from Site 696 record two distinct palaeofloras from the late Eocene Zone I to the early Oligocene Zone II assemblage that evolved in response to an increase in environmental disturbance beginning around 34.1 Ma. Throughout the Zone I assemblage (~37.6-33.6 Ma) abundant *Nothofagus* with secondary *Podocarpaceae*, minor angiosperm and cryptogam elements indicate the presence of a relatively humid *Nothofagus*-dominated temperate rainforest, growing under MATs between 10.1°C and 15.3°C, and MAP of 1499mm and 2043mm (Fig. 4). Comparison with lipid biomarker *n*-alkane results (Fig. 5.; López-Quirós et al., 2021) indicates ACL indicative of temperate vegetation, supporting this interpretation. In addition, marine palynomorphs (Houben et al., 2013) and calcareous nannofossil (Wei and Wise, 1990) assemblages attest to temperate marine depositional conditions. *Nothofagus* (predominantly *Fuscospora*-type), together with less common Podocarpaceae, formed the forest canopy across much of the mid- to higher-altitude areas, with tracts perhaps dominated by one or the other due to natural differences in shade tolerance (Poole, 1987; Veblen et al., 1996; Gallagher et al., 2008; Bowman et al., 2014). *Microcachrys* along with Araucariaceae, Ericaceae, Liliaceae, Chenopodioideae and low growing proteaceous shrubs, also reflect better drained higher-altitude habitats as well as coastal and marginal forest environments (Kühl et al., 2002; MacPhail et al., 1999; Kershaw and Wagstaff, 2001; Bowman et al., 2014). Prior to the opening of the Powell Basin the SOM was joined to the Antarctic Peninsula (King and Barker, 1988; López-Quirós et al., 2021), which was comparable in elevation to the Trans Antarctic Mountains and Dronning Maud Land during the late Eocene (Wilson et al., 2012). This may suggest that exposed parts of the SOM also had a similar mountainous elevation. Furthermore, the modern topography of the South Orkney Islands reaches a maximum of 1265m (~4150ft; USGS, 2021). Subsidence of the SOM since the late Eocene (López-Quirós et al., 2021), together with erosion likely mean these exposed parts of the SOM were once higher than today, supporting the reconstruction of higher and lower altitude vegetation communities. Today, similar cool temperate *Nothofagus*-dominated mixed-podocarp forests occur in the temperate Valdivian region of southern Chile, between 37°45' and 43°20'S (Veblen et al., 1983, 1996; Poole et al., 2001, 2003; Cantrill and Poole, 2012a; Bowman et al., 2014) across elevations greater than 2000m to lowland areas (Kershaw, 1988; Punyasena et al., 2011; Arias, 2015), where westerly trade winds from the Pacific result in high precipitation. Comparable mixed *Nothofagus*-podocarp forests are also found today in New Zealand (e.g., Wardle, 1984; Poole, 1987), however the geological setting of southern South America, with oceanic crust being subducted beneath a convergent continent margin, is most similar to that of the Antarctic Peninsula region during the Cenozoic (Poole et al., 2001; Cantrill and Poole, 2012a).

Pollen taxa representing vegetation communities with very different temperature requirements exist within the early-late Eocene (~35.5-37.6 Ma) Subzone Ia. The presence of the thermophilic taxa Arecaceae (palms), *Beauprea* and Myrtaceae, each occurring intermittently throughout this subzone (643.73-597.66 mbsf), indicates the existence of a temperate-thermophilic vegetation community. These communities are not recorded in coeval Antarctic Peninsula assemblages perhaps due to the

Antarctic peninsulas high mountainous palaeotopography (Wilson et al., 2012) and/or the lower latitude of the SOM. Thermophilic taxa likely occupied sheltered lowland areas and favourable coastal margins and would have required mild temperatures and the absence of winter frosts, owing to the frost sensitivity of extant palms (Larcher and Winter, 1981, Tomlinson, 2006, Eiserhardt et al., 2011, Reichgelt et al., 2018). Sporomorph based climate estimates reveal Coldest Month Mean Temperatures (CMMT) between 6.2°C and 11.9°C, well above freezing (Fig. 4). In addition, warmth-loving ferns including Gleicheniaceae and rare Davalliaceae also occur together with moisture-loving conifers such as *Dacrydium*, which only occur in Subzone Ia, and *Phyllocladus*, further indicating warm wet temperate conditions throughout this subzone. Today these taxa occur in subtropical to temperate regions in lowland sheltered environments, often thriving in wet humid conditions and severely disturbed or pioneer habitats at the margins of rainforests and waterways (Specht et al., 1992; Chinnock and Bell, 1998; Bowman et al., 2014; Arias, 2015). Similar conditions have also been documented in the late Eocene of southern New Zealand (Conran et al. 2016) suggesting a longitudinal continuum of relatively high precipitation and temperatures during this time interval.

The co-occurrence of prominent vegetation communities, each with very different temperature and moisture requirements therefore suggests that late Eocene forests across the northern Antarctic Peninsula and SOM were subject to climatic gradients related to differences in elevation and proximity to the coastline. Furthermore, the presence of thermophilic taxa within Subzone Ia and the lack of cold temperature taxa reveal conditions were warmer, by around 1.4°C between 37.6 and 35.5 Ma, compared to the rest of Zone I, indicating a phase of latest Eocene cooling from 35.5 to 35 Ma. The cooling between 35.5 and 35 Ma recorded by the terrestrial palynomorph assemblage coincides with a slight decrease in the terrestrial *n*-alkanes Index (TI; Mahiques et al., 2017), which records absolute input of *n*-$C_{27+29+30}$-rich molecules present in vascular plants, indicating decreased input of terrestrial plant-derived organic matter (Fig. 5; López-Quirós et al., 2021). The latest Eocene cooling recorded at Site 696 after 35.5 Ma corresponds with large-scale changes in vegetation composition and decreasing diversity from Antarctic Peninsula palaeoflora records (e.g., Askin, 2000; Anderson et al., 2011; Warny and Askin, 2011a, 2011b). Furthermore, an upwards-increase in illite clay minerals (Robert and Maillot, 1990) between approximately 36.4 to 33.9 Ma, signifying a shift in weathering regime from chemical to physical, supports the idea of latest Eocene climate cooling.

An initial spike in reworked Mesozoic sporomorphs at around 34.1 Ma at the onset of the EOT indicates an increase in reworking. This coincides with increasing Eocene dinoflagellate cyst taxa percentages over Protoperidiniaceae (Houben et al., 2013). Sediments within this EOT interval also exhibit two coarsening-upward packages, within an otherwise fine-grained sequence. Furthermore, these sedimentary packages contain the first evidence for ice-rafted debris (IRDs; Barker et al., 1988; López-Quirós et al., 2021) in conjunction with a high percentage of illite clay minerals (Robert and Maillot, 1990). Based on these observations, significant ice build-up around the northern Antarctic Peninsula and SOM is inferred during the latest Eocene, with a period of continental ice expansion to the coast or beyond (López-Quirós et al., 2021). This is supported by the presence of glacial surface textures on sand grains (Kirshner and Anderson, 2011) and rare drop stones (Wellner et al., 2011)

from late Eocene (34-37 Ma; Bohaty et al., 2011) sediments offshore James Ross Basin, as well as other sedimentological and geochemical evidence indicating late Eocene and early Oligocene cooling and ice expansion on the northern Antarctic Peninsula (e.g., Robert and Maillot, 1990; Ivany et al., 2008). Furthermore, topographic reconstructions indicate the northern Antarctic Peninsula at the EOT was comparable in elevation to the Trans Antarctic Mountains and Dronning Maud Land (Wilson et al., 2012). In model simulations these are suggested nucleation points for late Eocene and Oligocene glaciation (DeConto and Pollard 2003; DeConto et al., 2007), suggesting a glacial presence in the Antarctic Peninsula region is reasonable during this time (Carter et al., 2017; Lepp, 2018). However, the pollen and spore assemblage from site 696 as well as other late Eocene and Oligocene sporomorph assemblages from the Antarctic Peninsula (e.g., Anderson et al., 2011; Askin and Warny 2011a), indicate the region still retained some vegetation and therefore was never fully glaciated.

### 5.2.2 Early Oligocene Palaeoenvironment and Palaeoclaimte

Despite the evidence for a cooling of terrestrial climate between 35.5 Ma and 35 Ma, and latest Eocene glacial onset around 34.1 Ma the terrestrial palynomorph assemblage from Site 696 indicates that *Nothofagus*-dominated forests did not change dramatically in composition until the early Oligocene, after the EOIS. An expansion of conifer trees and cryptogams accompanied by a rapid increase in taxa diversity is recorded between approximately 33.5 and 32 Ma. This significant transformation of Antarctic flora in the early Oligocene is quantitatively reflected by the results of DCA analysis, as well as by a decrease in *n*-alkane ACL, which in turn suggest herbaceous plants and/or conifer dominance (Fig. 5; López-Quirós et al., 2021). The diversity patterns derived from the Shannon diversity index and rarefaction analysis show that early Oligocene forests contained a significantly higher number of taxa compared with the late Eocene forest communities. Previous studies (e.g., Cantrill, 2001; Raine and Askin, 2001; Prebble et al., 2006; Griener and Warny, 2015) suggest that significant cooling and drying of the climate across the EOT led to decreasing diversity and a lowering of the forest canopy, with low stature forests formed of scrubby dwarf *Nothofagus* and podocarps in sheltered regions and low altitude coastal zones. However, based on the relatively high diversity and frequency of cryptogam taxa and non-*Nothofagus* angiosperms within the Zone II assemblage, these *Nothofagus*-podocarp forests would have been at least intermediate in stature and relatively open, allowing the development of fern, shrub and bryophyte communities (e.g., Macphail and Truswell, 2004). When compared to *n*-alkane results, an increase in the abundance of cryptogams, specifically *Sphagnum* moss, is also supported by an increase in the aquatic plant *n*-alkane index ($P_{aq}$) and $n$-$C_{23}$/$n$-$C_{29}$ ratios throughout the early Oligocene (López-Quirós et al., 2021). *Sphagnum* moss has been found in many Antarctic Oligocene and Miocene assemblages interpreted to represent low diversity tundra vegetation (e.g., Rain, 1998; Askin, 2000; Askin and Raine, 2000; Prebble et al., 2006), suggesting a transition towards cooler tundra mosaic vegetation at this time (e.g., Truswell and Macphail, 2009). Furthermore, within the early Oligocene (~33.5-32.2 Ma) Zone II assemblage, the increase in *Coptospora* along with *Stereisporites* (*Sphagnum*), and some angiosperms such as *Liliacidites* and possibly *Myricipites*, likewise suggest the progression towards colder environments. *Coptospora* and *Liliacidites* have been found in Oligocene and Miocene assemblages across Antarctica, including the Ross Sea region (e.g., Askin and Rain, 2000; Prebble et al., 2006), Meyer Desert Formation (e.g., Lewis et al., 2008), Wilkes Land (e.g., Sangiorgi

et al., 2018) and the Antarctic Peninsula (e.g., Warny and Askin 2011a). These Oligocene/Miocene assemblages have been inferred to represent tundra mosaic vegetation in cold, possibly glacial, landscapes (Francis and Hill, 1996; Macphail and Truswell, 2004; Prebble et al., 2006). In addition, the presence of common millimetre sized IRDs between approximately 564 and 560 mbsf suggests phases of continental ice expansion to coastal and possibly shelf areas (Barker et al., 1988; López-Quirós et al., 2021). Within this same interval, sporomorph-based climate reconstructions also reveal a cooling and drying step, with a decrease in MAT and MAP from around 12.7°C to 10.5°C and 1895mm to 1471mm, between ~33.5 Ma and 33.4 Ma. The initially relatively high temperatures during the earliest Oligocene may be associated with the reported return to near-Eocene climate soon after the EOT (Liu et al., 2009; Houben et al., 2012; Wilson et al., 2013) and is possibly also reflected in a shift in some organic matter indices across the EOT and after the EOIS (Fig. 5; López-Quirós et al., 2021). However, further interpretation of this sediment section is hampered by a gap in core recovery. The increase in typical tundra taxa together with common IRD indicates a potential phase of cooling and glacial expansion during the earliest Oligocene, possibly punctuated by the development of milder climates (e.g., Liu et al., 2009; Houben et al., 2012; Wilson et al., 2013).

Warming and cooling phases with episodes of ice growth and retreat would have caused environmental disturbance, likely reducing the extent of Eocene *Nothofagus*-dominated forested. In New Zealand modern *Nothofagus* seedlings are able to capitalise on small canopy openings enabling them to out-compete podocarps in old-growth stands (Lusk et al., 2015). However, unlike podocarps, juvenile *Nothofagus* also require shelter from frost and desiccation, finding it hard to establish themselves amongst other vegetation in open and marginal forest environments until other vegetation has been partially suppressed through overtopping by *Nothofagus* (Wardle, 1964; Lusk et al., 2015 Rawlence et al., 2020). Thus, conifers are probably favoured by exogenous disturbance, providing a short-lived reprieve from angiosperm competition (Enright & Hill 1995; Lusk et al., 2015). Across the EOT and earliest Oligocene glacial related environmental disturbance and the development of milder climates, possibly after the EOT (e.g., Liu et al., 2009; Houben et al., 2012; Wilson et al., 2013), could have therefore facilitated the expansion of different vegetation types previously suppressed by the dominance of *Nothofagus*. Furthermore, a study by Galeotti et al. (2016), suggested that until ca. 32.2 Ma any Antarctic ice sheet would have been extremely sensitive to orbitally paced, local insolation forcing and would have been prone to large fluctuations. These fluctuations in ice extent would have in turn resulted in environmental disturbance. Variability in ice volume during the early Oligocene are also reflected by greatly varying benthic $\delta^{18}O$, confirming large fluxes in Antarctic ice during this time. Therefore, the unusual expansion of gymnosperms and cryptogams seen at Site 696 is suggested to be related to an increase in environmental disturbance caused by repeated glacial expansion and retreat, with the first major glacial expansion around 34.1 Ma, together with the competitive dominance of podocarps on exposed disturbed sites (Fig. 6). A lack of evidence for marine reworking after the EOT and good agreement between terrestrial biomarkers (López-Quirós et al., 2021) and the fossil sporomorph record suggest that the changes in the terrestrial palynomorph assemblage during the early Oligocene (~33.5 Ma) reflect true climate signals and increased environmental disturbance caused by glacial onset. Moreover, the results of this study reveal that major

changes in terrestrial vegetation took place after the onset of glaciation rather than after terrestrial climate cooling that took place during the latest Eocene after 35.5Ma.

## 5.3 Paleoceanography

Concurrent to terrestrial cooling at 35.5 Ma, indicated by the loss of thermophilic taxa and a decrease in sporomorph-based MAT estimates, dramatic changes to marine environments at site 696 are signalled by the appearance of glauconitic packstone (~588.8 to 577.9 mbsf; López-Quirós et al., 2019). The formation of this mature glaucony-bearing facies is suggested to be related to a decrease in the delivery of terrigenous sediments to Site 696 and suboxic reducing conditions at the sediment water interface (López-Quirós et al., 2019, 2021). This change in oceanic environmental conditions may be explained by the opening

of the proto-Powell Basin and changes to ocean currents, with several studies (e.g., Lawver and Gahagan, 1998; Eagles and Livermore, 2002; Livermore et al., 2007) indicating strengthening of Scotia Sea and the northern Weddell Sea circulation at this time (López-Quirós et al., 2021). Decreased terrigenous sediment supply, as the SOM moved away from the Antarctic Peninsula, is supported by a drop in sedimentation from ~4 cm/kyr between 645.6 to ~597.2 mbsf to ~1.85 cm/kyr between ~588.8 to 577.9 mbsf (López-Quirós et al., 2019, 2021). The opening of the Powell Basin to shallow and possibly intermediate

waters is also suggested to have resulted in the creation of an upwelling system fuelling high sea-surface primary productivity and the development of oxygen-deficient bottom waters (López-Quirós et al., 2021). Condensed glauconitic sections on outer shelf-upper slope setting commonly occur beneath upwelling areas with high productivity (e.g., Cook and Marshall, 1981; Wigley and Compton, 2006; Banerjee et al., 2016). Fe-enrichment of glaucony grains is likely the result of high sea-surface productivity as a result of upwelling along the margin of the SOM (López-Quirós et al., 2019, 2021). Further evidence for

increased marine biological productivity at Site 696 comes from distribution patterns of dinocysts and the proliferation in heterotrophic Protoperidiniaceae dinoflagellates, notably with increased abundances of the genera *Brigantedinium* spp., at approximately 34.5 Ma (Fig. 5; Houben et al., 2013, 2019). The dominance of Protoperidiniaceae dinoflagellates throughout the late Eocene-early Oligocene at Site 696 suggesting eutrophic surface waters supports the hypothesis of high sea-surface productivity enhanced by upwelling related to the opening of the Powell Basin at 35.5 Ma. Furthermore, high TOC within the

early Oligocene combined with the presence of pyrite and diagenetic barite provide further evidence suggesting high marine productivity, leading to low oxygen conditions at the seafloor, possibly due to upwelling (López-Quirós et al., 2021).

Importantly these change in oceanographic conditions associated with the opening of the Powell Basin occur synchronously with terrestrial cooling at 35.5 Ma, within the resolution of this study. Furthermore, large-scale changes in vegetation

composition and decreasing diversity from Antarctic Peninsula (e.g., Askin, 2000; Anderson et al., 2011; Warny and Askin, 2011a, 2011b) also occur at this time. Therefore, this may suggest a link between marine and terrestrial environments and that the opening of the Powell Basin and the establishment of oceanic upwelling may have driven a large-scale regional cooling step at 35.5 Ma. However, the regional change in oceanography and marine environments cannot be directly linked with the terrestrial vegetation change and glacial onset in the region that took place about one million years later, at 34.1 Ma. The timing

of the second cooling rather suggests that the event at site 696 is linked to global cooling at the onset of the EOT, documented by a combination of deep-ocean cooling and global ice sheet growth, marking the step from a largely ice-free greenhouse world to an icehouse climate (Hutchinson et al. 2020).

## 6. Conclusion

The terrestrial palynomorph assemblage from ODP Site 696 in the Weddell Sea records records palaeofloral evolution in
response to increased environmental disturbance and provide insight into late Eocene and early Oligocene terrestrial climate and cryosphere evolution. Late Eocene pollen and spore assemblages reveal a terrestrial climate cooling at 35.5 Ma with a decrease in MAT by an average of 1.4°C, associated with a shift from warm-temperate *Nothofagus*-dominated forests including typical thermophilic plant types to cool temperate *Nothofagus*-dominated forests. This cooling of terrestrial climate after 35.5 Ma coincides with changes in floral diversity and composition in palaeoflora records from the Antarctic Peninsula (e.g., Askin,
2000; Anderson et al., 2011; Warny and Askin, 2011a, 2011b), which have been interpreted to reflect the onset of prolonged cooling in the region. Despite evidence for terrestrial cooling and ice expansion, *Nothofagus*-dominated forests did not change dramatically in composition until the early Oligocene, when there was distinct expansion of gymnosperms and cryptogams accompanied by a rapid increase in taxa diversity between approximately 33.5 and 32 Ma. We suggest that glacial related environmental disturbance, starting around 34.1 Ma, reflected by an increase in cold climate taxa and sedimentological
evidence for ice transport and erosion (e.g., Robert and Maillot, 1990; López-Quirós et al., 2019, 2021), facilitated the expansion of different vegetation types previously suppressed by the dominance of *Nothofagus*.

The cooling step at 35.5 Ma coincides with an abrupt change to marine environments at Site 696, indicated by the appearance of mature glaucony-bearing facies (~588.8 to 577.9 mbsf; López-Quirós et al., 2019). Development of this glauconitic section
has been related to the opening of the Powell Basin, resulting in decreased sedimentation rates and the development of oceanic upwelling fuelling high marine biological productivity and the development suboxic bottom waters (López-Quirós et al., 2019, 2021). The coincidence between terrestrial cooling and changes to ocean currents and marine environments at Site 696 possibly indicates a strong link between ocean and terrestrial environmental change, suggesting the opening of the Powell Basin and reorganisation of ocean currents triggered a regional cooling step at 35.5 Ma prior to glacial onset at 34.1 Ma. However, the
large temporal gap (~1.4 Ma) between oceanographic changes and glacial onset suggests that the opening of ocean gateways did not alone trigger glaciation, even if ocean gateways may have played a role in stepwise cooling.

## Data Availability

All data will be available on the www.pangaea.de database (submitted 02/07/2021, awaiting validation)

**Acknowledgments**

NT received funding from the Natural Environment Research Council (NERC)-funded Doctoral Training Partnership ONE Planet [NE/S007512/1]. Funding for this research was also provided by the Spanish Ministry of Science and Innovation (grants CTM2014-60451-C2-1/2-P and CTM2017-89711-C2-1/2-P) cofounded by the European Union through FEDER funds. PKB acknowledges funding from the European Research Council for starting grant #802835, OceaNice. This work used Deep Sea
Drilling Project archived samples provided by the International Ocean Discovery Program (IODP). We thank the staff at the Gulf Coast core repository (GCR) for assistance in ODP Leg 113 core handling and shipping. We thank CNRS for the salary support of MAS.

**Author contributions**

NT and US designed the research and NT analysed pollen and spores. ALQ added sedimentological and geochemical analyses.
NT prepared the manuscript with contribution from all co-authors.

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

**Table captions**

Table 1: Revised age model for Ocean Drilling Program (ODP) Leg 113 Site determined by calcareous nannofossil and dinoflagellate cysts biostratigraphy (FO = First occurrence, FCO = Fist common occurrence)

Table 2: List of fossil pollen and spore taxa and their NLR used in sporomorph-based climate estimates from ODP Site 696.

**Figure captions**

Figure 1: Modern day geographical and tectonic setting of the study area, showing location of ODP Site 696 on the south-eastern margin of the SOM (red circle), and litho-tectonic units superimposed for the Antarctic Peninsula and southern South America (modified after Elliot, 1988). Tectonic setting and features after Maldonado et al. (2015). APR, Antarctic-Phoenix Ridge; BB, Bruce Bank; Sea; DB, Discovery Bank; DvB, Dove Basin; EB, Endurance Basin; ESR, East Scotia Ridge; FP, Falkland Plateau; HB, Herman Bank; JB, Jane Basin; JBk, Jane Bank; OB, Ona Basin; PB, Powell Basin; PBk, Protector Bank; PrB, Protector Basin; SB, Scan Basin; SGM, South Georgia Microcontinent; SI, Seymour Island; SOM, South Orkney Microcontinent; SSIB, South Shetland Islands Block; TR, Terror Rise; WSR, West Scotia Ridge; and WSS, West Scotia Sea. (Adapted from López-Quirós et al., 2019, 2021).

Figure 2: Stratigraphy of the studied sedimentary interval from ODP Site 696 Hole B. From left to right: Simplified lithological log of ODP Site 696, Age-depth plot based on biostratigraphy, cores, detailed lithological log of Eocene-Oligocene Unit VII and clay mineral percentage. Biostratigraphic age constraints based on calcareous nannofossils (Wei and Wise, 1990) and dinoflagellate cysts (Houben et al., 2013). Detailed lithological log from López-Quirós et al. (2019, 2021). Clay minerals are from Robert and Maillot (1990). (Adapted from López-Quirós et al., 2019, 2021).

Figure 3: Frequency and stratigraphic distribution of major pollen and spore taxa with CONISS ordination showing two distinct zones (Zone I and Zone II), Zone I further subdivided based on the occurrence of key taxa. Pollen and spore taxa have been separated into key ecological groups.

Figure 4: Sporomorph based quantitative climate estimates using probability density functions (PDF). From left to right: Coldest Month Mean Temperature (CMMT), Mean Annual Temperature (MAT), Warmest Month Mean Temperature (WMMT) and Mean Annual Precipitation (MAP).

Figure 5: Distribution of key vegetation and dinocyst groups plotted against diversity indices results, percentages of reworked terrestrial palynomorphs and *n*-alkane variables/ratios. From left to right: Rarefaction analysis results at number of species per 50 and 100 specimens, percentage of reworked terrestrial palynomorphs, DCA axis 1 results, percentage of cryptogam taxa, percentage of angiosperm taxa (non-*Nothofagus*), percentage of gymnosperm taxa, percentage of *Nothofagus*, percentage of endemic-Antarctic dinocyst taxa, percentage of Protoperidiniaceae dinoflagellates, TOC, TI index, ACL, Paq and n-alkane n-C23/n-C29 ratios. Dinoflagellate abundance from Houben et al. (2013). Terrestrial biomarkers/n-alkane variables/ratios from López-Quirós et al. (2021).

Figure 6: Schematic representation of vegetation from Site 696, illustrating the response of key taxonomic group to climate and environmental change through key intervals during the late Eocene and early Oligocene. (A) Late Eocene vegetation (~37.6-35.5 Ma), during the deposition of Zone Ia vegetation was dominated by *Nothofagus* with secondary podocarps and an understory of cryptogams and minor angiosperms. Sporomorph-based climate estimates and the presence of thermophilic taxa indicate conditions were relatively warm compared to the rest of the section; (B) Latest Eocene after late Eocene climate cooling (~35.5-34.1 Ma), vegetation remained similar to that of the late Eocene and was still dominated by *Nothofagus*, but climate cooling by around 2°C resulted in loss of thermophilic taxa and slight decrease in taxa diversity; (C) EOT vegetation during glacial onset (~34.1 Ma), environmental disturbance caused by ice expansion and retreat resulted in the reduction of *Nothofagus*-dominated forested areas and increase in tundra-like vegetation; (D) early Oligocene (~33.5-32.2 Ma), during glacial retreat and the development of milder climates disturbance and reduction of *Nothofagus*-dominated forests facilitated the expansion of more competitive Podocarpaceae and pioneer cryptogam taxa.

**Table 1**

| Event/Characteristic | Kind | Lower level | Upper level | Bottom depth | Top depth | Mid-depth | Reference | Age (Ma) | Reference |
|---|---|---|---|---|---|---|---|---|---|
| *FO Chiropteridium galea* | Dinocysts | 53R-3, 80 cm | 53R-2, 130 cm | 552.70 | 551.70 | 552.20 | Houben et al., 2019 | <33.26 | Pross et al., 2010 |
| *FO Malvinia escutiana* | Dinocysts | 55R-1, 117 cm | 55R-1, 62 cm | 569.39 | 568.82 | 569.11 | Houben et al., 2013; 2019 | 33.6 | Houben et al., 2011 |
| *FO Stoveracysta kakanuiensis* | Dinocysts | 55R-3, 75 cm | 55R-2, 147 cm | 571.95 | 571.16 | 571.55 | Houben et al., 2013; 2019 | 34.1 | Clowes, 1985 |
| *FO Reticulofenestra oamaruensis* | Calcareous nannofossils | 58R-1, 122cm | 57R-1, 112 cm | 598.42 | 588.72 | 593.57 | Wei and Wise, 1990 | 35.5 | Villa et al., 2008 |
| *FCO Isthmolithus recurvus* | Calcareous nannofossils | 60R-1, 36 cm | 59R-CC | 616.96 | 616.6 | 616.78 | Wei and Wise, 1990 | 36.27 | Villa et al., 2008 |
| *FO Reticulofenestra bisecta* | Calcareous nannofossils | 62R-6, 132 cm | - | 643.62 | - | 643.62 | Wei and Wise, 1990 | <37.61 | Villa et al., 2008 |

**Table 2**

| Fossil taxa | Botanical affinity | Reference | NLR used for climate analysis |
|---|---|---|---|
| **Angiosperms** | | | |
| *Acaena* sp. | *Acaena* | | *Acaena* |
| *Arecipites* sp. | Arecaceae | Raine et al. (2011) | Arecaceae |
| *Beaupreaidites cf. verrucosus* | Proteaceae (*Beauprea*). | Raine et al. (2011) | *Beauprea* |
| *Chenopodipollis chenopodiaceoides* | Amaranthaceae (Chenopodioideae) | Raine et al. (2011) | Chenopodiaceae (Chenopodioideae) |
| *Clavatipollenites ascarinoides* | Chloranthaceae (*Ascarina*). | Barreda et al. (2020) | *Ascarina* |
| *Cupanieidites orthoteichus* | Sapindaceae | Raine et al. (2011) | Sapindaceae |
| *Ericipites cf. scabratus* | Ericaceae | Raine et al. (2011) | Ericaceae |
| *Lateropora glabra* | Pandanaceae (*Freycinetia*) | Raine et al. (2011) | *Freycinetia* |
| *Liliacidites intermedius* | Liliaceae (?*Arthropodium*) | Raine et al. (2011) | Liliaceae |
| *Lymingtonia cf. cenozoica* | Nyctaginaceae (*Pisonia brunoniana*) | Raine et al. (2011) | *Pisonia* |
| *Malvacipollis cf. subtilis* | Malvaceae?/Euphorbiaceae | Raine et al. (2011) | Euphorbiaceae |
| *Myricipites harrisii* | Casuarinaceae/Myricaceae | Raine et al. (2011) | Myricaceae |
| *Myrtaceidites cf. mesonesus* | Myrtaceae (*Metrosideros*) | Barreda et al. (2020, 2021) | *Metrosideros* |
| *Nothofagidites asperus* complex | *Nothofagus* (*Lophozonia*) | Raine et al. (2011) | *Nothofagus menziesii* |
| *Nothofagidites brachyspinulosus* | *Nothofagus* (*Fuscospora*) | Raine et al. (2011) | *Fuscospora* |
| *Nothofagidites lachlaniae* | *Nothofagus* (*Fuscospora*) | Raine et al. (2011) | *Fuscospora* |
| *Nothofagidites emaricidus* complex | *Nothofagus* (*Brassospora*) | Raine et al. (2011) | *Brassospora* |
| *Nothofagidites flemingii* | *Nothofagus* (*Nothofagus*) | Barreda et al. (2020, 2021) | *Nothofagus* |
| *Nothofagidites roccanensis* | *Nothofagus* (*Nothofagus*) | Barreda et al. (2021) | *Nothofagus* |
| *Nothofagidites* spp. | Nothofagaceae | Barreda et al. (2020, 2021) | Nothofagaceae |
| *Propylipollis reticuloscabratus* | Proteaceae (*Gevuina/Hicksbeachia*) | Barreda et al. (2020, 2021) | Proteaceae |
| *Proteacidites* spp. | Proteaceae | Barreda et al. (2020, 2021) | Proteaceae |
| *Sparganiaceaepollenites barungensis* | Sparganiaceae (*Sparganium*) | Macphail & Cantrill (2006) | *Sparganium* |
| **Gymnosperms** | | | |
| *Araucariacites australlis* | Araucariaceae (*Araucaria*) | Barreda et al. (2020, 2021) | *Araucaria* |
| *Dacrydiumites praecupressinoides* | Podocarpaceae (*Dacrydium cupressinum*) | Raine et al. (2011) | *Dacrydium* |
| *Microalatidites paleogenicus* | Podocarpaceae (*Phyllocladus*) | Barreda et al. (2021) | *Phyllocladus* |
| *Microcachryidites antarcticus* | Podocarpaceae (*Microcachrys tetragona*) | Barreda et al. (2020, 2021) | Podocarpaceae |
| *Phyllocladidites mawsonii* | Podocarpaceae (*Lagarostrobos franklinii*). | Barreda et al. (2021) | *Lagarostrobos franklinii* |
| *Podocarpidites* spp. | Podocarpaceae (*Podocarpus*) | Barreda et al. (2020, 2021) | *Podocarpus* |
| *Podosporites* spp. | Podocarpaceae (cf. *Microcachrys*) | Raine et al. (2011) | Podocarpaceae |
| *Trichotomosulcites subgranulatus* | Podocarpaceae (*Microcachrys*) | Barreda et al. (2021) | Podocarpaceae |
| **Cryptogams** | | | |
| *Baculatisporites comaumensis* | Osmundaceae | Barreda et al. (2020) | Osmundaceae |
| *Ceratosporites cf. equalis* | Selaginellaceae (*Selaginella*) | Raine et al. (2011) | Selaginellaceae |
| *Coptospora archangelskyi* | Bartramiaceae (*Conostomum*) | Raine (1998) | *Conostomum* |
| *Cyathidites* spp. | Cyatheaceae/ Dicksoniaceae/ Schizaeaceae | Barreda et al. (2020) | Cyatheaceae |
| *Dictyophyllidites arcuatus* | Gleicheniaceae (?*Dicranopteris*) | Raine et al. (2011) | *Dicranopteris* |
| *Foveotriletes lacunosus* | Lycopodiaceae (*Huperzia*) | Raine et al. (2011) | *Huperzia* |
| *Gleicheniidites* spp. | Gleicheniaceae | Barreda et al. (2020) | Gleicheniaceae |
| *Laevigatosporites* spp. | Polypodiaceae | Barreda et al. (2020) | Polypodiaceae |
| *Monolites alveolatus* | cf. Polypodiaceae (*Belvisia*) | Raine et al. (2011) | *Belvisia* |
| *Osmundacidites cf. wellmanii* | Osmundaceae (*Todea barbara*) | Raine et al. (2011) | Osmundaceae |
| *Polypodiisporites cf. radiatus* | Davalliaceae (*Davallia*) | Conran et al. (2010) | *Davallia* |
| *Retitriletes/Lycopodiacidites* | Lycopodiaceae (*Lycopodium*) | Raine et al. (2011) | *Lycopodium* |
| *Stereisporites* spp. | Sphagnaceae (*Sphagnum*) | Truswell & Macphail (2009) | *Sphagnum* |

1105

1110    **Figure 1**

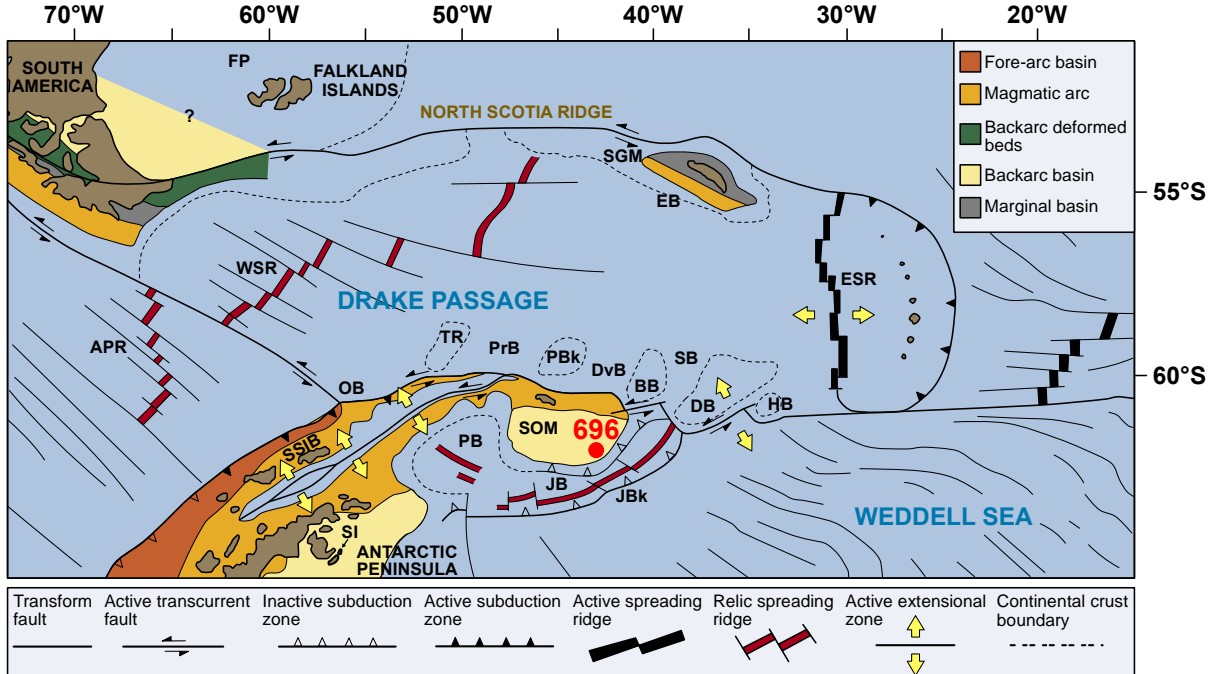

**Figure 2**

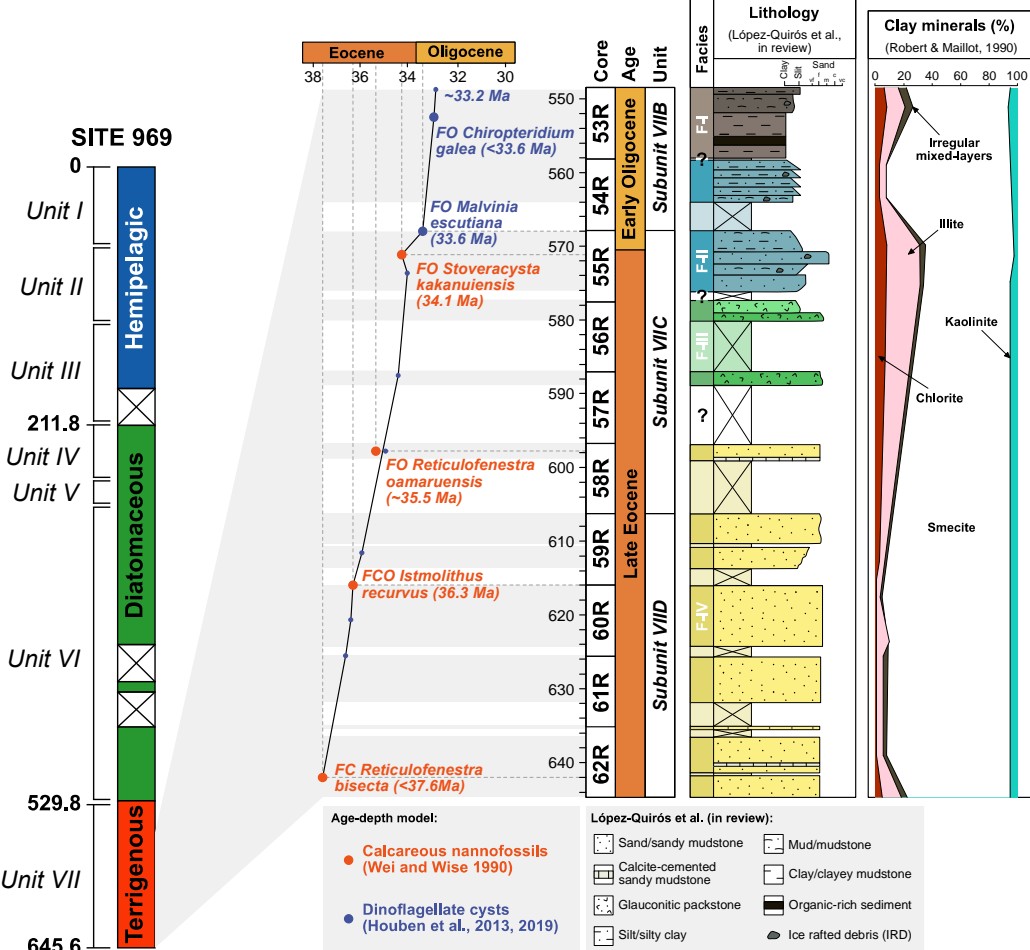

**Figure 3**

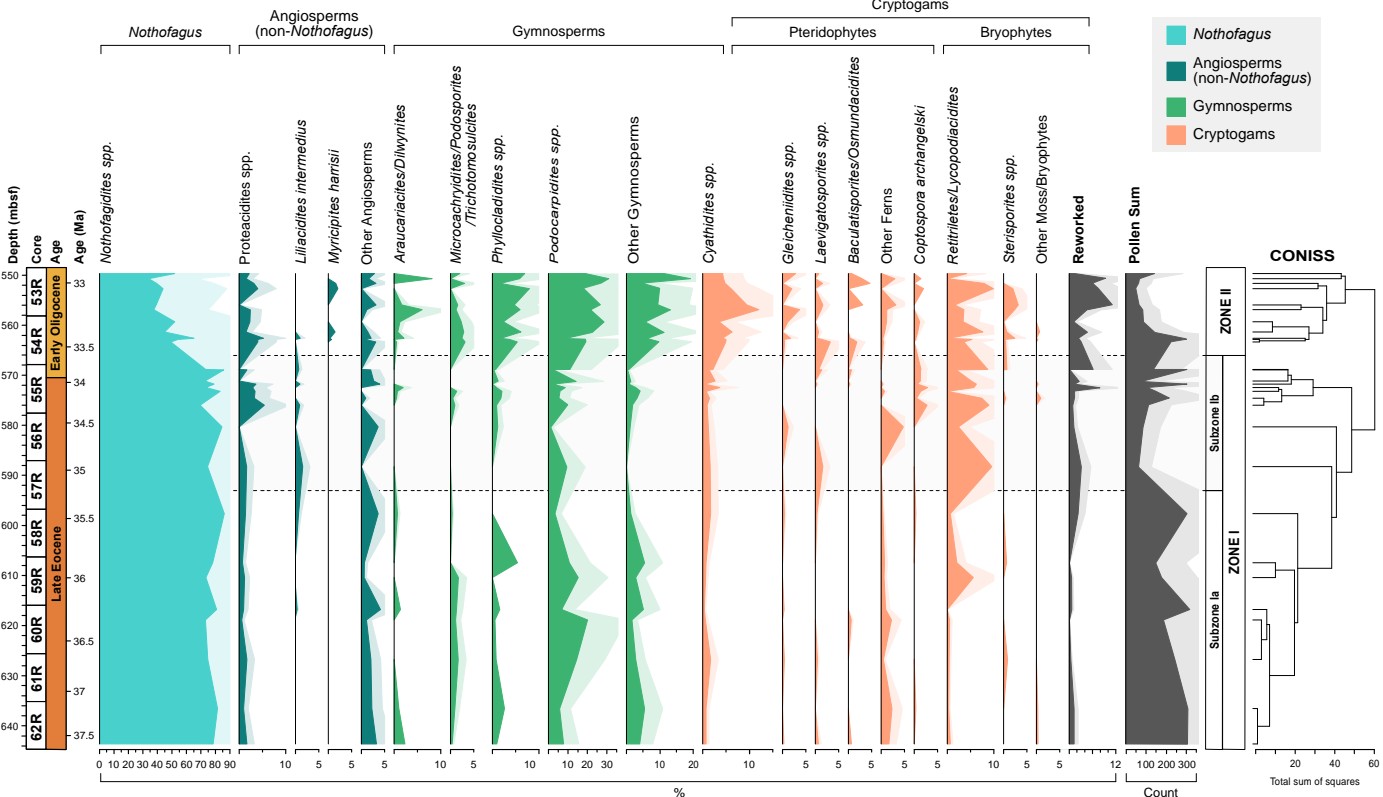

**Figure 4**

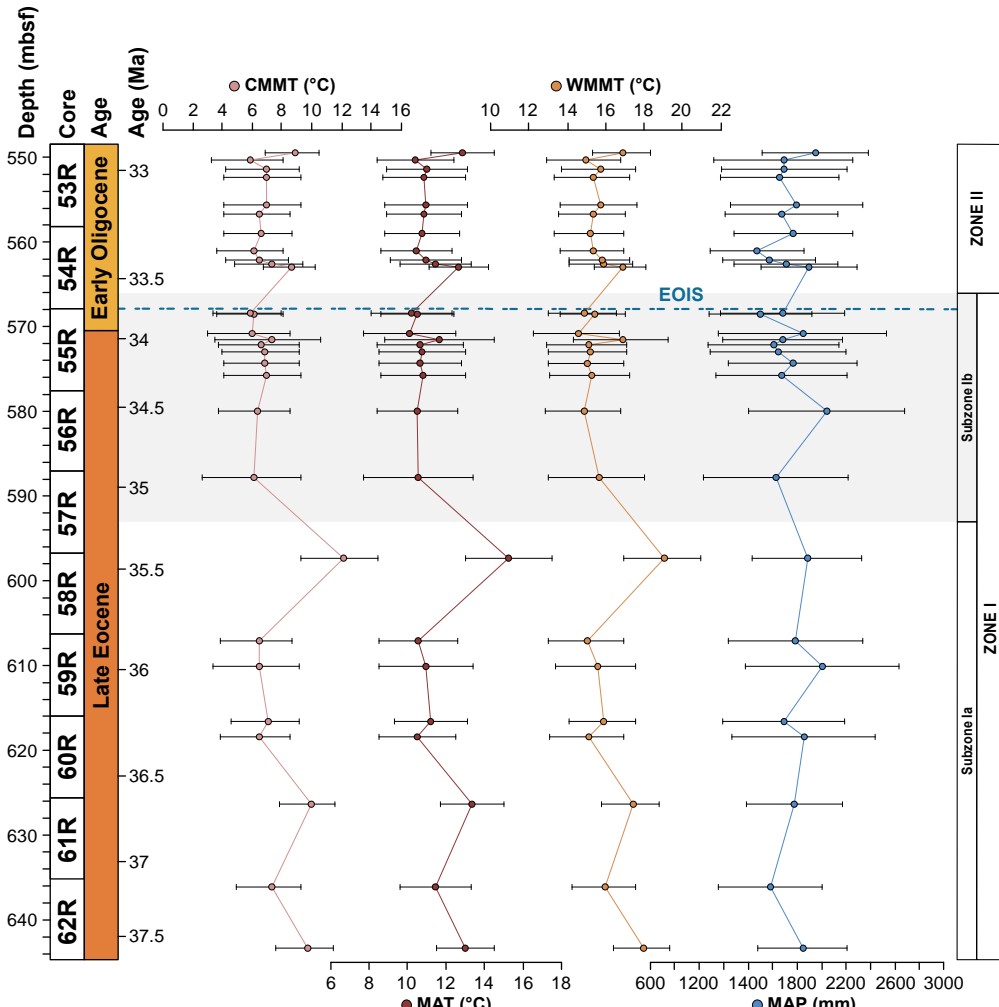

1155

1160

**Figure 5**

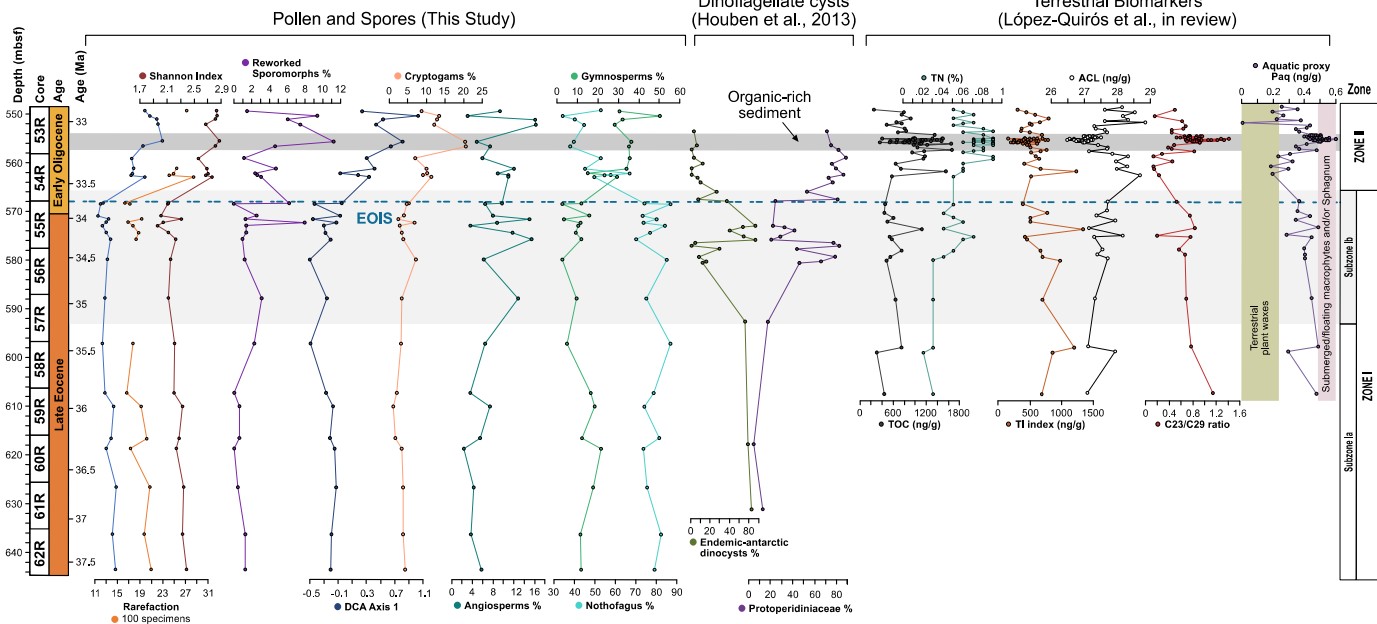

**Figure 6**

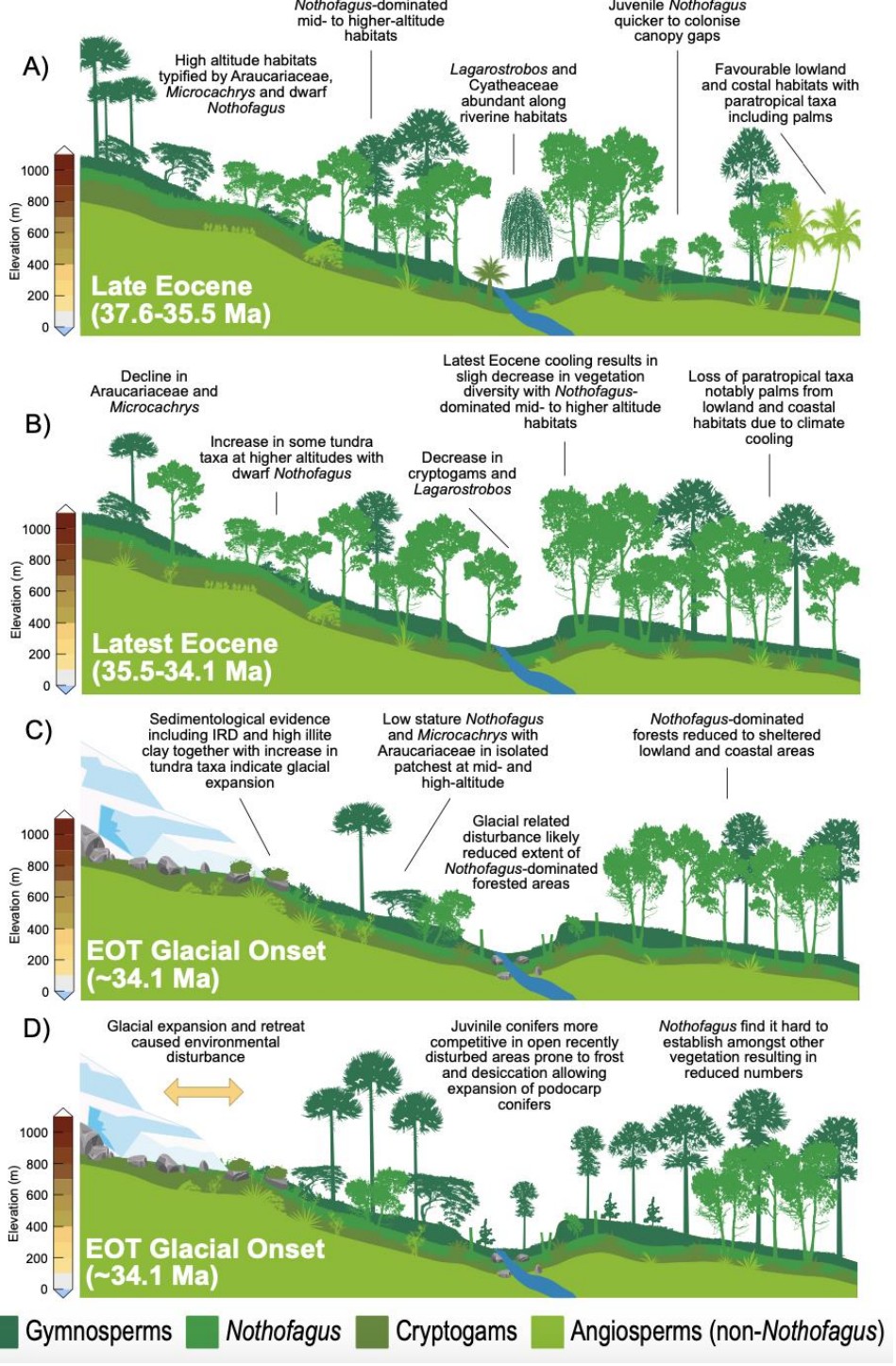