# Peer review of "Vegetation change across the Drake Passage region linked to late Eocene cooling and glacial disturbance after the Eocene-Oligocene Transition"

_Climate of the Past, 2021_

## Referee Comment (RC2)

[Figure]

**Figure 1**

[Figure]

[Figure]

**Figure 2**

[Figure]

[Figure]

**Figure 3**

[Figure]

[Figure]

**Figure 4**

[Figure]

[Figure]

**Figure 5**

[Figure]

[Figure]

[Figure]

**Figure 6**

[Figure]

[referee-annotated manuscript omitted]

---

## Author Response (AR1)

Dear referee and editorial team,

Please find below our response to the comments the reviewer rose. We thank the reviewers for constructive and positive feedback on our manuscript. We propose the changes indicated in the text below.

Kind regards

Nick Thompson, behalf of all co-authors.

Response to Reviewer #1

REV#1 General Comment:
The authors present a palynological record from the South Orkney microcontinent in the Wedell Sea and associated environmental changes using nearest living relative-based paleoclimate reconstructions, lipid biomarker geochemistry and sedimentological changes. They identify a cooling step in the late Eocene and a vegetation turnover around the EOT boundary, which they associate with regional tectonic changes leading to oceanographic changes. The study is well-conceived and provides some unique data, since obtaining high-resolution and well-resolved terrestrial climate information from southern South America sheds light on the complex opening of the Drake Passage, specifically of the Powell Basin. Opening of the Drake Passage has long been postulated as a possible cause for global cooling at the EOT boundary (as well as the Oligocene-Miocene boundary), but this study can show that there is an offset between the timing of global glaciation at the EOT and regional vegetation and oceanographic reorganization associated with opening of the Powell Basin. I do not see any major flaws in the conceptualization of this study, nor with the conclusions that the authors draw. The results are novel and shed light on an important geological event. The figures are stellar. I suggest minor revisions based on two changes that the authors may want to consider, and one additional problem.

**Authors response:** We are pleased to hear such positive feedback and we thank the referee for these constructive comments and will respond in detail below.

REV#1 Suggestion 1:
Table 2 contains the pollen/spore types and the associated botanical affinity. Most of the botanical affinities are based on Raine et al. (2011) and some on various other references. Raine et al. (2011) is an excellent resource and there is a strong biogeographical connection between New Zealand, Antarctica and southern South America. Still, Raine et al. (2011) base their nearest living relatives predominantly on associations made in New Zealand and to a lesser extent Australia and Antarctica. It might therefore be prudent for the authors to confirm botanical affinities as applied in South American studies, such as those by Viviana Barreda, since this is likely an important floral source at SOM. I put some references that the authors can use for this in my minor comments below.

**Authors response:** We revisited each of the Nearest Living Relatives (NLR) for fossil taxa used in our palaeoclimate estimates using the references provided (e.g., Barreda et al., 2020, 2021). Many of the taxa retained the same NLR with only a couple having different NLR.

**Proposed changes:** Table 2 has been amended to include references for Barreda (2020, 2021) and any new botanic affiliations. The new botanic affiliations are as follows: (1) *Laevigatosporites spp.* = Blechnaceae (Rain et al., 2011) -> *Laevigatosporites spp.* = Polypodiaceae (Barreda et al., 2020); (2) *Nothofagidites flemingii* = *Nothofagus* subg. *Fuscospora* (Rain et al., 2011) -> *Nothofagidites flemingii* = *Nothofagus* subg. *Nothofagus* (Barreda et al., 2020, 2021). Where NLR taxa were the

same between Raine et al. (2011) and Barreda et al., (2020, 2021), the latter reference was used, to highlight associations in South American studies. New climate estimate calculations using the new NLR taxa were also carried out and graphs and figures adjusted accordingly. The new climate estimate results do not affect the overall trends previously observed or any of the conclusions drawn. For new temperature and precipitation estimates please see Figure 4.

REV#1 Suggestion 2:
The *Nothofagus* subgenus *Brassospora* (or Nothofagaceae genus Trisyngyne, if you want to follow Heenan & Smissen 2013: Revised circumscription of *Nothofagus* and recognition of the segregate genera *Fuscospora*, *Lophozonia*, and *Trisyngyne* (Nothofagaceae), Phytotaxa, 146) is not used separately in the nearest living relative based paleoclimate reconstructions. The reason cited is its questionable range in New Caledonia. I've added some literature in my minor comments below to the research on Brassospora in New Guinea, where it is native and has quite a large range and is not hampered by geographic restrictions that an island such as New Caledonia poses. A possible concern with Nothofagaceae pollen could be that they dominate any assemblage, warm or cold, and thereby homogenize any climate signal that may be obtained from these records. This is valid, considering that Nothofagaceae pollen travel far and wide beyond their place of origin. I put a recommendation in for that as well (applying an abundance threshold). Additionally, Araucariaceae does not seem to be included in the nearest living relative analyses either (at least it's not in Table 2). If it wasn't, then it probably should be (Araucaria/Agathis), if it was an oversight in Table 2, then the authors should revise Table 2.

**Authors response:** We agree with the reviewers' comments and suggestion, and we have revisited the use of each of the *Nothofagus* subgenera. However, we have not used the circumscripture of Heenan and Smissen (2013) and prefer to keep using the palynologically long-established and well described subgenera *Brassospora*. In addition, we have revisited Araucariaceae, which was previously omitted due to the suggested NLR (*Araucaria Araucana*; Bowmann et al., 2014) having to few occurrences within the GBIF database and agree with the inclusion of *Araucaria* (Barreda et al. 2020, 2021) for the fossil taxa *Araucariacites australlis*. The concern with Nothofagaceae pollen dominating assemblages homogenize climate and ecological patterns was discussed. Our results clearly show a decrease in Nothofagaceae pollen and an associated increase in Podocarpaceae pollen not recorded in coeval Antarctic Peninsula or South American assemblages. In our opinion this suggests a true climate and/or ecological signal and that, despite Nothofagaceae pollen dominance throughout the studied section, this signal has not been masked or homogenized.

**Proposed changes:** Table 2 has been amended to include each of the *Nothofagus* subgenera and *Araucaria* (sensu Barreda et al., 2020, 2021) now used in palaeoclimate estimates. New climate estimate calculations using the new NLR taxa were carried out and graphs and figures adjusted accordingly. The new climate estimate results do not affect the overall trends previously observed or any of the conclusions drawn. For new temperature and precipitation estimates please see Figure 4.

REV#1 Problem:
A recurring reference involved in interpretations of the results is a reference to López-Quirós et al. (in review): Eocene-Oligocene paleoenvironmental changes in the South Orkney Microcontinent (Antarctica) linked to the opening of Powell Basin. Considering the importance of these results (sedimentology & organic geochemistry) in the interpretation of the opening of the Powell Basin, it seems prudent to await acceptance/publication of that paper before this paper is accepted and published.

**Authors response:** The study by López-Quirós et al. (2021) was an important reference that was used at aid and compliment the results of this study. At the time of writing López-Quirós et al. was

still awaiting publication. However, I am pleased to say it has since been accepted and published in the journal Global and Planetary Change, Volume 204.

**Proposed changes:** The citation has been updated in the bibliography and in text.

REV#1 Specific comment, lines 76-77:
"Where possible … and evaluation." For nearest living relative techniques, you can in addition to establishing relative abundances using 300 counts, account for rare occurrences by scanning the entire slide without counting. Was this done?

**Authors response:** Scanning for rare occurrences was not conducted. We feel that rare taxa if present would not add any more detail to environmental or paleoclimate interpretations. Very rare taxa are also often a limited value for regional environmental reconstructions as they may represent long distance pollen transport or reworking. We are convinced that the taxa that were identified already presented a robust picture of palaeo-environment and climate, together with interpretations from sedimentology and geochemical biomarkers.

**Proposed changes:** None.

REV#1 Specific comment, lines 272:
"also continue to … unable to be properly identified." If it is not possible to identify these taxa, wouldn't there be a risk that these are reworked Mesozoic taxa? Undifferentiated bisaccate grains, including pollen that resemble *Podocarpidites*, are common in the Mesozoic.

**Authors response:** We agree with this comment and have revisited the identification of "unidentified bisaccate" taxa. The group is not reworked as indicated by their level of preservation and thermal maturity, when compared with reworked taxa from this study. We rather grouped under "unidentified bisaccate" all *Podocarpidites* that could not be further differentiated. To avoid confusion, we merged this group now into "*Podocarpidites spp.*"

**Proposed changes:** Renaming Unidentified bisaccates to *Podocarpidites spp.* The renaming has not affected the results of this study, or the conclusions drawn.

REV#1 In text corrections:
The reviewer has proposed a number of smaller in text corrections and changes.

**Authors response:** We agree with the corrections proposed by the reviewer.

**Proposed changes**: Changes made in text per reviewers' recommendations.

REV#1 Figure 1:
very nice figure! I will only point out that on my screen the "North Scotia Ridge" was hard to read. Perhaps make it a somewhat darker hue (or black?).

**Authors response:** We agree with the alteration proposed by the reviewer.

**Proposed changes:** Font colour changed to a darker colour. Please see figure 1.

REV#1 Table 2:
Raine et al. 2011 is a great resource for nearest living relatives, but it's also somewhat risky to apply widely in the Southern Hemisphere, because the nearest living relatives in this database are

primarily (though not solely) established based on New Zealand pollen types. There is a lot of biogeographic overlap with southern South America and New Zealand at the Eo/Oligocene boundary. Still, I suggest confirming the appropriateness of these nearest living relative assignments with the South American literature. Primarily Viviana Barreda's papers and Luis Palazzesi. See for example Table 1 in Barreda et al. 2020: Early Eocene spore and pollen assemblages from the Laguna del Hunco fossil lake beds, Patagonia, Argentina. International Journal of Plant Sciences 181. Or Table 1 in Barreda et al. 2021: The Gondwanan heritage of Eocene – Miocene Patagonian floras. Journal of South American Earth Sciences 107.

Table 2 appears to exclude Araucariaceae. Was this family not included in NLR analysis? I understand for *Dilwynites*, as it has a strongly relict distribution. However, if the family was excluded for some reason, this should probably be stated in the methodology.

**Authors response:** This issue has been addressed in suggestions 1 and 2. Please see above.

**Proposed changes:** Table 2 has been amended to include references for Barreda (2020, 2021) and any new botanic affiliations. The new botanic affiliations are as follows: (1) *Laevigatosporites spp.* = Blechnaceae (Rain et al., 2011) -> *Laevigatosporites spp.* = Polypodiaceae (Barreda et al., 2020); (2) *Nothofagidites flemingii = Nothofagus* subg. *Fuscospora* (Rain et al., 2011) -> *Nothofagidites flemingii = Nothofagus* subg. *Nothofagus* (Barreda et al., 2020, 2021). Where NLR taxa were the same between Rain et al., (2011) and Barreda et al., (2020, 2021), the latter reference was used, to highlight associations in South American studies. *Araucaria* (sensu Barreda et al., 2020, 2021) now used in palaeoclimate estimates and presented in table two. New climate estimate calculations using the new NLR taxa were carried out and graphs and figures adjusted accordingly. The new climate estimate results do not affect the overall trends previously observed or any of the conclusions drawn. For new temperature and precipitation estimates please see Figure 4.

Response to Reviewer #2

Rev#2 General comment:
The manuscript presents new terrestrial paleoclimatic data from high southern latitudes. A strength of the manuscript is that it presents and evaluates both palynofloral and geochemical proxies for paleoclimatic change across the EOT. The authors conclude that climate cooling began at 35.5 Ma, coinciding with the opening of the Powell Basin and importantly, prior to glacial onset at 34.1 Ma. The palynofloral evidence is well presented, but the geochemical (n-alkane and TOC) methods and results need reorganizing. One concern is that the authors refer to high-altitude habitats in the text and in Figure 6. I would like them to clarify whether there is tectonic evidence of this or whether it is inferred from the palynofloral assemblages? Although a number of these taxa do inhabit montane areas today because that is where precipitation is highest, throughout the Cenozoic they are known to have proliferated in wet, low-lying areas throughout the southern hemisphere (eg. SE Australia, New Zealand). I suggest the authors carefully reconsider this interpretation. Notably, the wet conditions inferred at the study site might ensure that these plant taxa could have inhabited low land areas. Overall, this study is an excellent contribution to our understanding of southern hemisphere palynofloras and paleoclimates.

**Authors response:** We thank the reviewer for their constructive comments and suggestions and are pleased to hear such positive feedback. We agree that interpretations of altitude based on taxa alone needs more information to support the suggestion of higher and lower altitude communities. There is evidence that the Antarctic Peninsula was comparable in elevation to the Trans Antarctic Mountains and Dronning Maud Land during the late Eocene (Wilson et al., 2012). Prior to the opening of the Powell Basin the South Orkney Microcontinent (SOM) was still joined to the Antarctic

Peninsula (King and Barker, 1988; López-Quirós et al., 2021). This may suggest that exposed parts of the SOM also had a similar elevation. Subsidence of the SOM related to the opening of the Powell Basin (López-Quirós et al., 2021) and erosion have likely reduced the high of these exposed parts of the SOM since the late Eocene. Today topography of the South Orkney Islands reaches a maximum of 1265m (~4150ft; USGS, 2021). We therefore suggest it was likely that there were areas of the SOM that were at higher elevations and supported higher altitude vegetation.

**Proposed changes:** Lines 371-375: "Prior to the opening of the Powell Basin the SOM was joined to the Antarctic Peninsula (King and Barker, 1988; López-Quirós et al., 2021), which was comparable in elevation to the Trans Antarctic Mountains and Dronning Maud Land during the late Eocene (Wilson et al., 2012). This may suggest that exposed parts of the SOM also had a similar mountainous elevation. Furthermore, the modern topography of the South Orkney Islands reaches a maximum of 1265m (~4150ft; USGS, 2021). Subsidence of the SOM since the late Eocene (López-Quirós et al., 2021), together with erosion likely mean these exposed parts of the SOM were once higher than today, supporting the reconstruction of higher and lower altitude vegetation communities."

Rev#2 Specific comments:
In section 2.3 (Materials and Methods) the authors comment that "The following section will focus on the interpretation of lipid biomarker (n-alkane) and stable isotope data from Site 696". However, I encourage the authors to remove most of this section as it contains background information and therefore seems out of place. Most of the text in sections 2.3.1 and 2.3.2 should be removed and instead incorporated into the discussion or included as background information earlier in the text, not in the materials and methods section. Please only include the equations you need for the results/discussion.

All results in sections 2.3.1 and 2.3.2 should be moved to the results section and placed under relevant subheadings. Please differentiate palynofloral results, n-alkane results and TOC results if the former two are indeed new to this study. If the n-alkane and TOC results are being reporting in Lopez-Quiros et al., in review then please do not report them here and instead refer to them in the discussion. For example, in the results you would state that TOC increased and, in the discussion, list the possible reasons why. This section should outline how you derived these results.

**Authors response:** The n-alkane and TOC results have been very recently published in López-Quirós et al. (2021) and we therefore agree with Rev#2 suggestions. However, we believe these sections provide useful information to the reader on the background of the geochemical analysis that was previously carried out as well as illustrating some key trends and patters in the geochemical data that can and have been used to aid interpretations made in this study. Therefore, we propose the inclusion of Section 2. "Previous Geochemical Analyses" as part of the Introduction between intro and methods

**Proposed changes:** We have added the additional section "Previous Geochemical Analyses", in order to display the geochemical results and provide the reader with background information (Lines 62-125). Following Rev#2 suggestions, we have also deleted the geochemical paragraphs in the Method section and removed all equations for geochemical analyses.

Rev#2 Specific comments:
For section 3.1 I recommend separating this into two paragraphs, with each clearly distinguishing the differences between Subzone 1a and Subzone 1b. Place the MAT and MAP results at the end of each paragraph too.

Authors response: We agree the inclusion of subzones within a single section may have been confusing and we have amended this section to make our results clearer.

Proposed changes: Lines 237-265 have been restructured following Rev#2 suggestions.

Rev#2 Specific in text comments:
Lines 204-205. Can the authors please clarify why all of these weren't identified to species level?

**Authors response:** Previously taxa listed in Lines 204-205 (now lines 205-209) were grouped into genera, e.g., all *Nothofagus* taxa identified to species level grouped under the genus *Nothofagus* spp. and so on. We agree that this dilutes the information available to the reader and have rewritten these lines to include species names.

**Proposed changes:** Lines 205-209: Pollen affiliated with the modern-day genus *Nothofagus* are the most abundant throughout the section, with pollen taxa belonging to the *Nothofagidites lachlaniae* complex, undifferentiated *Nothofagidites spp., Nothofagidites rocaensis* and the *Nothofagidites brachyspinulosus* complex being the largest groups. Other major pollen and spore taxa, in order of decreasing abundance include, undifferentiated *Podocarpidites spp.,* undifferentiated *Retitrileties/Lycopodiacidites spp.*, *Podocarpidites cf. exiguus*, pollen belonging to the *Podocarpidites marwickii/ellipticus* complex, *Cyathidites minor* and *Phyllocladidites mawsonii*, which occur commonly throughout the Eocene and Oligocene sections.

Rev#2 Specific in text comments:
Lines 215-216. Can the authors please explain why the rarefraction analysis was based on 50 specimens?

**Authors response:** There may have been some confusion here. Only samples containing 50 or more in situ sporomorph grains were used, not 50 specimens. We have amended this in text. Furthermore, we selected samples that contained a count of 50 or more grains to be used in analysis because the trends and patters reveals were similar to those if only using samples that contained 100 or more grains. The use of samples that contained 50 or more grains allowed us to include more samples and fill in some of the gaps left throughout the section if only using samples that contained 100 or more grains. In addition, we have run statistical evaluation of our count sizes based on the methods outlined in Djamali, M. and Cilleros, K., (2020). Statistically significant minimum pollen count in Quaternary pollen analysis; the case of pollen-rich lake sediments. Review of Palaeobotany and Palynology, 275, p.104156. Using the Pearson correlation coefficient, we achieved a mean of 0.86 (SD 0.1). These values can be considered excellent and provides confidence in our results.

**Proposed changes:**
Lines 83-84 "Only samples containing 50 or more in situ sporomorph grains were used for further analysis and evaluation."

Line 100 "Samples containing less than 50 grains were omitted from this analysis."

Lines 172-173 ". Based on rarefaction analysis, the average number of sporomorph species per sample is 13.28 ± 1.05 (mean ± SD) at a count of 50 grains."

Lines 223-224 "Based on the results of rarefaction analysis the average number of sporomorph species for a count size of 50 grains is 19.63 ± 2.00.

Rev#2 Specific in text comments:

Please elaborate on any other possible palynofloral sources.

**Authors response:** Per the reviewers' recommendations we have elaborated on possible sources.

**Proposed changes:** Lines 310-321: "Sediments may also have been supplied from the southern tip of South America (e.g., the Magallanes Basin and the Fuegian Andes; Carter et al., 2017), due to the more proximal location of the SOM to South America prior to its separation from Antarctica during the Eocene (Eagles and Jokat, 2014). However, detrital zircon ages clearly show a strong dissimilarity between Site 696 samples and South America (Carter et al., 2017). Furthermore, the occurrence of well-preserved palynomorphs and moderate to well-preserved in situ benthic foraminifera, with predominantly angular to subangular terrigenous particles, does not support the notion of long-distance transport of sediments from adjacent sources (e.g., Seymour Island and southern South America; López-Quirós et al., 2021). These observations, together with an expansion of gymnosperm conifers and cryptogams recorded during the early Oligocene (33.5-32.2 Ma) at Site 696, but absent from Antarctic Peninsula and southern South America floras (e.g., Askin et al., 1992; Anderson et al., 2011), suggest that the vegetation of the SOM was unique in character. It is therefore likely that a significant proportion of detrital material, including sporomorphs, was likely of local origin (e.g., exposed parts of the SOM), with some input from the northern Antarctic Peninsula and possibly southern South America during the late Eocene."

Rev#2 Specific in text comments:
Please outline what the significant differences are between SOM and Antarctic Peninsula palynofloras.

**Authors response:** There may have been some confusion with this point. The differences between the SOM and Antarctic Peninsula are explained in the following sentences (lines 321-327): "In agreement with previous observations by Mohr (1990) the sporomorph assemblage from Site 696 contains a greater diversity of angiosperm pollen compared to late Eocene Antarctic Peninsula palaeofloras (e.g., Anderson et al., 2011; Warny and Askin 2011b; Warny et al., 2019). This higher diversity has also been reported in southern South American Paleogene sporomorph floras (e.g., Romero and Zamaloa, 1985; Romero and Castro, 1986). In addition, the late Eocene Zone Ia assemblage (37.6-35.5 Ma) at Site 696 contains the thermophilic taxa *Arecipites spp.* (Arecaceae), *Myrtaceidites cf. mesonesus* (Myrtaceae), and *Polypodiisporites cf. radiatus* (*Davallia*) not recorded in coeval Antarctic Peninsula assemblages, possibly due to the more northern latitude of the SOM resulting in milder climatic conditions."

**Proposed changes:** Minor alterations to the wording of this section to make the comparison between these two regions clearer.

Rev#2 Specific in text comments:
Line 293. Please elaborate on the precipitation requirements of the taxa too, as opposed to focussing on temperature alone.

**Authors response:** The line in question (now lines 307-310) is within section 5.1 "Sediment Transport and Provenance". We therefore believe that specific references to climate requirements of vegetation are not required in this section. However, we have taken the reviwers' comments onboard and added clearer links to vegetation, precipitation in modern habitats and palaeoclimate estimates.

**Proposed changes:** Minor alterations to the wording within section 5.2 "Palaeoenvironment and Palaeoclimate" to elaborate on the precipitation requirements of the taxa and the precipitation regime certain taxa suggest with links to palaeoclimate estimates.

Rev#2 Specific in text comments:
Lines 316-322. Could the authors explain why they dismiss having more than one source area (i.e., one local and one more regional).

**Authors response:** Per the reviewers' recommendation we have added more information regarding sediment source. We maintain our suggestion that the majority of sediment was supplied from exposed parts of the SOM based on the differences outlined in the pollen and spore assemblage between the SOM and adjoining regions (e.g., South American and the Antarctic Peninsula). We do accept that some sediment was likely supplied from some of these adjacent areas and have indicated accordingly intext.

**Proposed changes:** Lines 339-358: ". Barriers to the delivery of sediment by long distance gravity flows from the margins of the southern Weddell Sea, further suggested that sediments may have been transported to the SOM by icebergs (Carter et al., 2017). In spite of this, the presence of in situ thermophilic taxa within the early-late Eocene of Site 696 (37.6-35.5 Ma) suggests mild and even ice-free conditions during this overlapping time period. Furthermore, palaeo-sea-surface temperature reconstructions (Douglas et al., 2014) indicate relatively warm conditions (~14°C), and fossil dinoflagellate cyst (Houben et al., 2013, 2019), calcareous nannofossils (Wei and Wise, 1990) and smectite-dominated clay mineralogy (Fig. 2: Robert and Maillot, 1990) support temperate depositional conditions (López-Quirós et al., 2021) not favourable for transport by ice. Unequivocal evidence for ice transport, in the form of ice-rafted debris, at Site 696 is observed within two coarse-grained mudstone intervals within a fine-grained transgressive sequence deposited around 34.1 Ma (Barker et al., 1988; López-Quirós et al., 2021). However, these intervals contain altered glaucony grains most likely sourced from shallower SOM coastal/shelf areas (López-Quirós et al., 2019, 2021). Therefore, these observations and those of this study suggest that transportation by ice from adjacent land areas (e.g., Antarctic Peninsula and Ellsworth–Whitmore Mountains) was unlikely before 34.1 Ma and that a majority of sediments transported to Site 696 are likely of local origin from exposed parts of the SOM as the Powell basin opened isolating the microcontinent from the possible sediment supply of the Antarctic Peninsula and southern Weddell Sea margins."

Rev#2 Figure 3:
Please include a column for the autochthonous pollen sum on Figure 3.

**Authors response:** We agree with the correction.

**Proposed changes:** Autochthonous pollen sum column added. Please see Figure 3.

Rev#2 Technical corrections
The reviewer has proposed a number of smaller in text corrections and changes.

**Authors response:** We agree with the corrections proposed by the reviewer.

**Proposed changes:** Changes made in text per reviewers' recommendations